# How to build a ribosome from RNA fragments in *Chlamydomonas* mitochondria

Florent Waltz [1,2,3,8], Thalia Salinas-Giegé [2,8], Robert Englmeier[4], Herrade Meichel[2], Heddy Soufari[1], Lauriane Kuhn [5], Stefan Pfeffer [6], Friedrich Förster [4], Benjamin D. Engel [3,7], Philippe Giegé [2✉], Laurence Drouard [2✉] & Yaser Hashem [1✉]

Mitochondria are the powerhouse of eukaryotic cells. They possess their own gene expression machineries where highly divergent and specialized ribosomes, named hereafter mitoribosomes, translate the few essential messenger RNAs still encoded by mitochondrial genomes. Here, we present a biochemical and structural characterization of the mitoribosome in the model green alga *Chlamydomonas reinhardtii*, as well as a functional study of some of its specific components. Single particle cryo-electron microscopy resolves how the *Chlamydomonas* mitoribosome is assembled from 13 rRNA fragments encoded by separate non-contiguous gene pieces. Additional proteins, mainly OPR, PPR and mTERF helical repeat proteins, are found in *Chlamydomonas* mitoribosome, revealing the structure of an OPR protein in complex with its RNA binding partner. Targeted amiRNA silencing indicates that these ribosomal proteins are required for mitoribosome integrity. Finally, we use cryo-electron tomography to show that *Chlamydomonas* mitoribosomes are attached to the inner mitochondrial membrane via two contact points mediated by *Chlamydomonas*-specific proteins. Our study expands our understanding of mitoribosome diversity and the various strategies these specialized molecular machines adopt for membrane tethering.

[1] Institut Européen de Chimie et Biologie, U1212 Inserm, Université de Bordeaux, 2 rue R. Escarpit, 33600 Pessac, France. [2] Institut de biologie moléculaire des plantes, CNRS, Université de Strasbourg, 12 rue du général Zimmer, 67084 Strasbourg, France. [3] Helmholtz Pioneer Campus, Helmholtz Zentrum München, Ingolstädter Landstraße 1, 85764 Neuherberg, Germany. [4] Structural Biochemistry, Bijvoet Centre for Biomolecular Research, Utrecht University, Universiteitsweg 99, 3584 CG Utrecht, The Netherlands. [5] Plateforme protéomique Strasbourg Esplanade FRC1589 du CNRS, Université de Strasbourg, 67084 Strasbourg, France. [6] Zentrum für Molekulare Biologie der Universität Heidelberg, DKFZ-ZMBH Alliance, Im Neuenheimer Feld 282, 69120 Heidelberg, Germany. [7] Department of Chemistry, Technical University of Munich, Lichtenbergstraße 4, 85748 Garching, Germany. [8] These authors contributed equally: Florent Waltz, Thalia Salinas-Giegé. ✉email: philippe.giege@ibmp-cnrs.unistra.fr; laurence.drouard@ibmp-cnrs.unistra.fr; yaser.hashem@inserm.fr

Mitochondria are essential organelles of eukaryotic cells that act as metabolic hubs and powerhouses, producing energy through aerobic respiration. They still possess their own genome and gene expression machineries, vestiges of their once free-living bacterium ancestor[1,2]. Due to the evolutionary drift of eukaryotes, mitochondrial complexes involved in metabolism and gene expression combine features from their bacterial ancestor with traits that evolved in eukaryotes[3,4]. The final step of gene expression, translation, is carried out by specialized mitochondrial ribosomes (mitoribosomes). They synthesize the few proteins still encoded by the mitochondrial genome, most of which are hydrophobic components of the respiratory chain. Despite their shared prokaryotic origin[5], mitoribosome structure and composition were shown to be highly divergent across eukaryotes. They systematically acquired numerous additional ribosomal proteins (r-proteins), and their ribosomal RNAs (rRNAs) were either greatly reduced, like in animals and kinetoplastids[6–9], or expanded, like in plants and fungi[10–13].

Among the most prominent unresolved questions in the mitochondria biology field is the long-standing debate over the peculiar organization of the mitochondrial genome in the unicellular green alga *Chlamydomonas reinhardtii* and the biogenesis of its mitoribosome. This organism is widely used to study photosynthesis and cilia[14], but it is also an excellent model to investigate mitochondrial biology. It is one of the few organisms where mitochondrial transformation is possible[15], and mitochondrial mutants are viable in photoautotrophic conditions[16]. In contrast to vascular plants, or Viridiplantae in general, which are characterized by gene-rich and largely expanded mitochondrial (mt)-genomes, *C. reinhardtii* possess a small linear mt-genome of 16 kb. It only encodes eight proteins (all membrane-embedded components of the respiratory chain), three transfer RNAs (tRNAs) and, most intriguingly, non-contiguous pieces of the large subunit (LSU) and small subunit (SSU) ribosomal RNAs (rRNAs), scrambled across the genome[17–19]. The mt-genome is transcribed as two polycistronic primary transcripts synthesized from opposite strands[17,19]. Individual transcripts are then generated from the primary transcripts to produce mature, functional RNAs. When initially characterized 30 years ago[17], *Chlamydomonas* mitoribosome rRNA fragmentation represented the earliest example that an rRNA does not need to be continuous in order to be functional[20]. Although it was predicted that the rRNA fragments would somehow be integrated into a functional ribosome[18], it is enigmatic how these fragments are recruited, interact with each other, and are stabilized to form the 3D mitoribosome structure.

Here, we combine cryo-electron microscopy (cryo-EM) with in situ cryo-electron tomography (cryo-ET) to resolve the structure of a green algal mitoribosome, stunningly different from both its prokaryotic ancestor, as well as from the flowering plant mitoribosome[11], but also from all other characterized mitoribosomes across diverse species[3]. Our structure reveals how the reduced and fragmented rRNAs are organized and stabilized in the mitoribosome via numerous *Chlamydomonas*-specific r-proteins. Cryo-ET resolves the native structure and organization of *Chlamydomonas* mitoribosomes inside mitochondria, revealing that these mitoribosomes are exclusively bound to the inner mitochondrial membrane. Our study provides an example of a mitoribosome composed of numerous rRNA fragments, revealing a strikingly divergent blueprint for building this conserved molecular machine.

## Results

### Isolation, mass spectrometry, and cryo-EM of mitoribosomes.
To analyze the *C. reinhardtii* mitoribosome, mitochondria were purified and used for mitoribosome isolation following a procedure based on sucrose density gradient separation (see Methods)

(Supplementary Fig. 1). Collected fractions were systematically analyzed by nano-LC MS/MS (Supplementary Table 1) and screened by cryo-EM to determine their composition. This approach allowed us to identify fractions containing the two mitoribosome subunits, which were subsequently used for data collection (Supplementary Fig. 1). Proteomic analysis identified putative *Chlamydomonas*-specific r-proteins that were then confirmed by the corresponding cryo-EM reconstructions. Following image processing and extensive particle sorting, reconstructions of both dissociated subunits were obtained. The large subunit (LSU) was resolved to 2.9 Å, while the small subunit (SSU) was reconstructed at 5.49 Å and further refined to 4.19 Å for the body and 4.47 Å for the head using a focused refinement approach (Supplementary Fig. 2). Fully assembled mitoribosomes were identified by nano-LC MS/MS in the cytoribosome fraction (Supplementary Fig. 1), but cryo-EM investigation revealed aggregates in this fraction, most likely corresponding to mitoribosomes. Nevertheless, the individual subunit reconstructions were docked into the map of the entire *C. reinhardtti* mitoribosome obtained from subtomogram averaging of the in situ cryo-ET data (see below), allowing accurate positioning of both subunits relative to each other in the context of a fully assembled native mitoribosome. The isolated subunit reconstructions were similar to the in situ subtomogram average, demonstrating that they represent the mature LSU and SSU and not assembly intermediates. Notably, all densities corresponding to *Chlamydomonas*-specific r-proteins were present in both single particle and subtomogram average reconstructions.

### Overall structure of the *Chlamydomonas* mitoribosome.
Our cryo-EM reconstructions, along with our extensive MS/MS analyses, allowed us to build atomic models of both *C. reinhardtii* mitoribosome subunits (see "Methods" section) (Figs. 1 and 2). The overall architecture of this mitoribosome (Fig. 1) is clearly distinct from both its bacterial ancestor and the flowering plant mitoribosome[11]. *Chlamydomonas*-specific proteins and domains largely reshape both subunits. Similar to all previously described mitoribosomes, the *Chlamydomonas* mitoribosome has more r-proteins compared to its bacterial counterpart[3,4]. These proteins include ancestral r-proteins conserved with bacteria, mitoribosome-specific r-proteins shared with other mitoribosomes, and *Chlamydomonas*-specific r-proteins. In total, the *Chlamydomonas* mitoribosome contains 47 r-proteins in the LSU and 36 in the SSU. These include 11 new r-proteins (not accounting for unknown densities), 8 in the LSU and 3 in the SSU. The total of 83 r-proteins greatly exceeds the 54 r-proteins in bacterial ribosomes (Fig. 2 and Supplementary Table 1). As a result, very few rRNAs are exposed to the solvent, with proteins coating the entire mitoribosome and stabilizing the fragmented rRNAs (Supplementary Movie 1). Proteins follow the classical r-protein nomenclature[21], and newly identified proteins are numbered according to the last inventory of mitoribosomal r-proteins[22].

Reconstruction of the LSU (Fig. 1d) revealed eight additional r-proteins, named mL113 to mL119, plus PPR*, a putative PentatricoPeptide Repeat (PPR) protein (Figs. 1, 2 and Supplementary Fig. 4, Supplementary Table 1). They are distributed across the whole LSU, where they extend into the solvent and are anchored to the ribosome by interacting with both conserved r-proteins and rRNA fragments. With the exception of mL119 at the exit of the peptide channel, all these proteins are relatively large RNA binders composed of repeated alpha-helical folds, including a mitochondrial TERmination Factor (mTERF) protein, several OctotricoPeptide Repeat (OPR) proteins, and PPR*.

The small subunit (Fig. 1c) reconstruction highlights several distinctive features. Most strikingly, the SSU is shaped by two

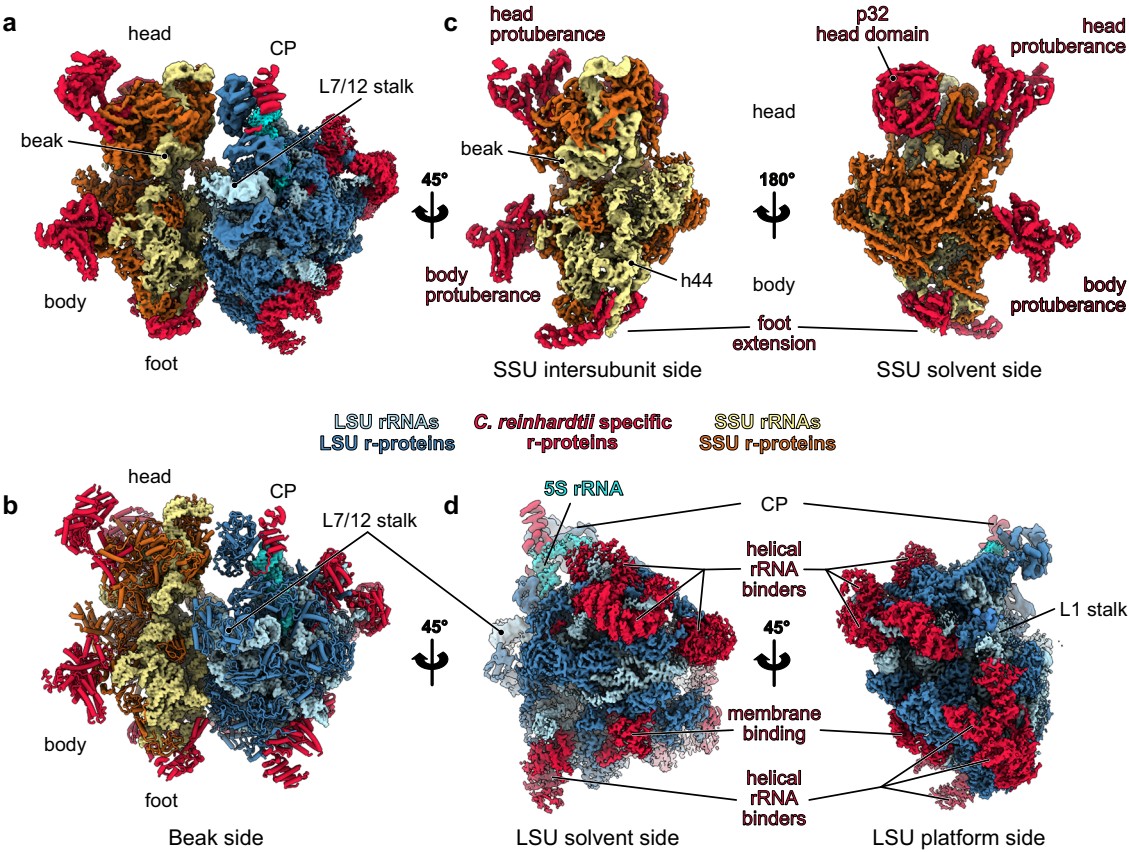

**Fig. 1 Overall structure of the *Chlamydomonas* mitochondrial ribosome. a** Composite cryo-EM map of the *Chlamydomonas reinhardtii* mitochondrial ribosome and **b** the resulting atomic model. The large subunit (LSU) components are depicted in blue shades, the small subunit (SSU) components are shown in yellow shades, and the specific r-proteins and domains are displayed in red. **c, d** Different views of the cryo-EM reconstructions of the SSU (**c**) and the LSU (**d**).

large protuberances positioned on its beak side, one on the head and one on the body, both formed by helical-rich proteins. The body protuberance, located close to the mRNA entrance, is mainly formed by a specific extension of more than 400 aa in the mitochondria-specific r-protein mS45 (Supplementary Fig. 5), a highly variable r-protein[22]. The head protuberance density could not be assigned due to the low resolution of this area. However, several conserved proteins of the SSU head located nearby, notably uS3m, uS10m, and mS35, present large extensions. Therefore, it is likely that these extensions could come together and form the head protuberance, as no other apparent candidates could be identified by MS/MS analyses. On the solvent side of the head, a large torus-shaped domain protrudes in the solvent. This additional domain is a homotrimeric complex formed by three copies of mS105, also called p32 (Supplementary Fig. 5a). This MAM33-family protein is seemingly conserved in all eukaryotes and has been described to have several functions in mitochondria, some related to mitoribosome assembly[23–26]. However, mS105/p32 was never before reported as a core component of a ribosome. Additionally, the head of the SSU is characterized by its missing beak, which is typically formed by helix 33 at the junction site of rRNA fragments S3 and S4 (Fig. 3c and Supplementary Fig. 7). The foot of the SSU is reshaped by *Chlamydomonas*-specific r-proteins. The extension is formed by two super-helical proteins, one PPR (mS106) and one OPR (mS107) identified by MS/MS and confirmed by AlphaFold[27]. The mS106 protein occupies a position similar to mS27 in humans[6,7] and fungi[12], but it does not appear to interact with RNA, nor does it share any sequence identity with mS27 (Supplementary Fig. 5d). On the

other hand, the OPR mS107 directly interacts with rRNA fragment S2, where it encapsulates the tip of helix 11 (Supplementary Fig. 5c).

**Fragmented ribosomal RNAs are assembled to reconstitute the core of *Chlamydomonas* mitoribosome.** In contrast to flowering plants, where rRNAs are largely expanded, the *C. reinhardtii* mitoribosome is characterized by its reduced and fragmented rRNAs (Fig. 3). These rRNAs are scrambled in the mitochondrial genome (Fig. 3a), where they are expressed as a single polycistron that is then further processed into matured transcripts by currently unknown endonucleases[17]. The "23S" and "16S" rRNAs are respectively split into eight fragments totaling 2035 nt, and four fragments totaling 1200 nt (Fig. 3c and Supplementary Figs. 6, 7, Supplementary Supplementary Movie 1). This corresponds to 30% and 22% reductions compared to bacteria (Fig. 3f). Among all the rRNA pieces predicted to be integrated into the mature mitoribosome, all but one (L2b) could be identified in our cryo-EM reconstructions. To confirm the absence of the L2b fragment, we performed comparative RNAseq analyses of mitochondrial and mitoribosomal fractions (Fig. 3b). Consistently, all rRNA fragments could be identified in the mitoribosome fraction except L2b. However, this analysis confirmed that the fragment is indeed expressed and accumulates in the purified mitochondria fraction (Fig. 3b), which is in line with previous transcriptomic analyses[19,28,29]. These results were further confirmed by RNA blots hybridized against L2b and an L2a control found in the ribosome and stabilized by the r-protein mL116 (Fig. 3d). Therefore, the L2b RNA is not associated with the mitoribosome,

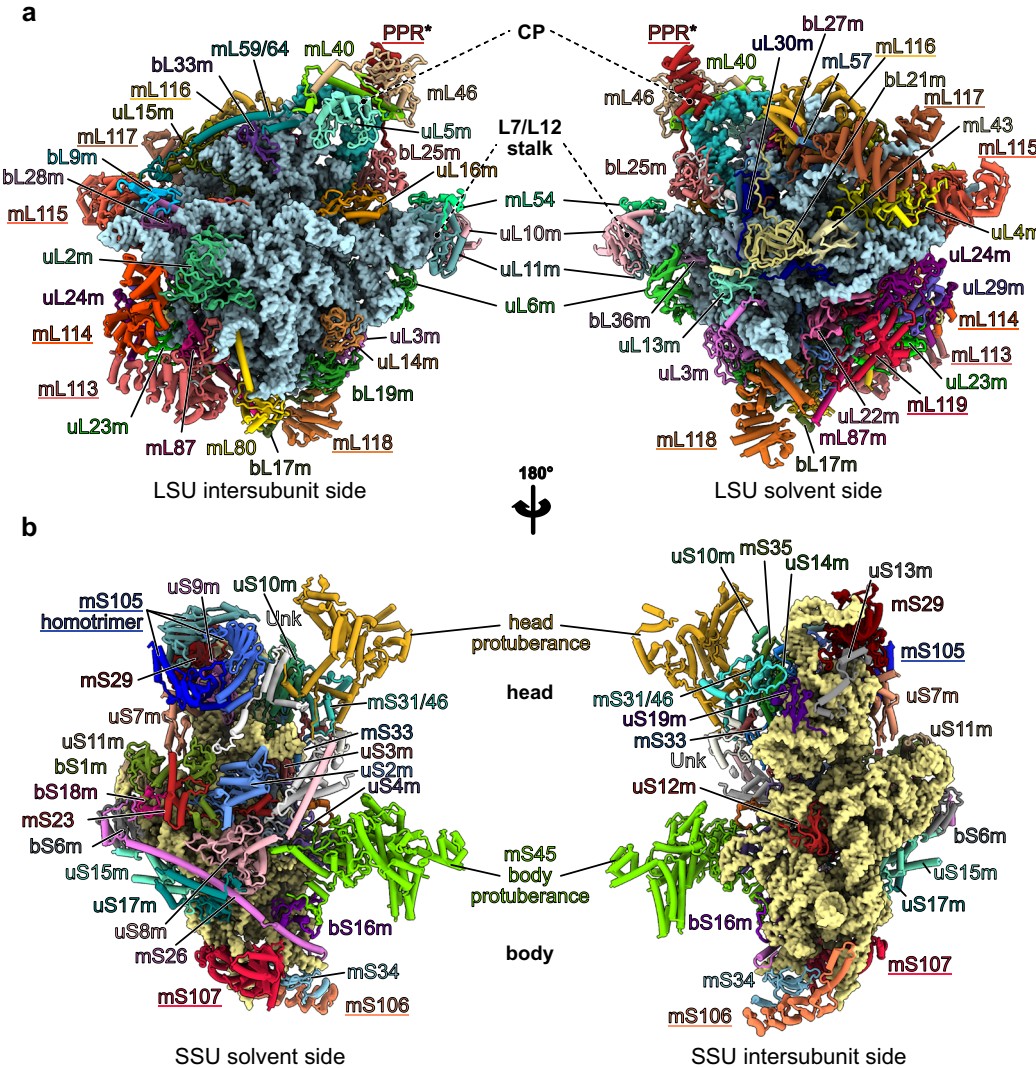

**Fig. 2 Ribosomal proteins of the *Chlamydomonas* mitoribosome. a, b** The atomic models of the *Chlamydomonas* mitoribosome LSU (**a**) and SSU (**b**), with mitoribosomal proteins shown in cartoon representation and individually colored and annotated. rRNAs are shown in surface representation and colored in light blue (LSU) and beige (SSU). Newly identified r-proteins are highlighted with underlined names.

suggesting that this small RNA has an independent function that remains to be elucidated.

In the SSU, the fragmented rRNAs form only a few interactions with the additional r-proteins and are mainly stabilized by base-pairing with each other (with the exception the S2 fragment's h11), which is encapsulated by mS107 (Supplementary Figs. 5 and 7). Fragments S1, S2, and a small portion of S3 form the 5′ domain, with the rest of S3 making up domain C. The 3′ end of S3 and the entirety of S4 constitute domains 3′M and 3′m, with S4 largely contributing to linking the head and body of the SSU. The region most conserved with bacteria is the decoding center, made by h1-2 and h27-28 (Fig. 3 and Supplementary Fig. 7).

In contrast to the SSU, the nine rRNA fragments in the LSU are all stabilized by the newly identified *Chlamydomonas*-specific r-proteins. These fragments reconstitute the different domains of the large subunit. L1 forms the highly reduced domain I of the LSU. L2a, L3b, L4, L5, and part of L6 together form domain II. Portions of L6 and L7 form the highly reduced—almost deleted—domain III. Fragments L7 and L8, the largest of all, make up domains IV, V, and VI, which form the catalytic core of the ribosome. These three domains are the least altered, with only a few helices missing and two expansion segments ES-66 and ES-94 (Fig. 4 and Supplementary

Fig. 6). The peptidyl transferase center (PTC) formed by H89 to H93 is particularly conserved with bacteria. The conservation of these domains is most likely due to the high selective pressure to conserve the catalytic region of the ribosome[30]. These rRNA fragments are held together by base-pairing with each other, and their extremities are stabilized by base-pairing with other fragments, e.g., L2a, L3b, and L4 (Fig. 3e) or with themselves. These results corroborate the initial predictions made 30 years ago by Boer and Gray[17]. However, several single-stranded rRNA extremities are also stabilized by the *Chlamydomonas*-specific r-proteins (see below). Surprisingly, while it was anticipated that 5S rRNA should be absent from *Chlamydomonas* mitoribosome, we identified an RNA density at the typical position of the 5S rRNA in the central protuberance (CP) (Fig. 1 and Supplementary Fig. 8). This rRNA density could be attributed to the L3a rRNA fragment (Supplementary Fig. 8). We further confirmed its association with the mitoribosome by comparative RNAseq analysis of mitochondrial and mitoribosomal fractions (Fig. 3b). Previous studies of the *C. reinhardtii* mitochondrial genome and rRNAs always failed to identify this rRNA as a putative 5S[17–19,31]. While L3a likely derives from an ancestral bacterial 5S rRNA, it has highly diverged; very little of the primary sequence is conserved with other 5S rRNAs (Supplementary Fig. 8d),

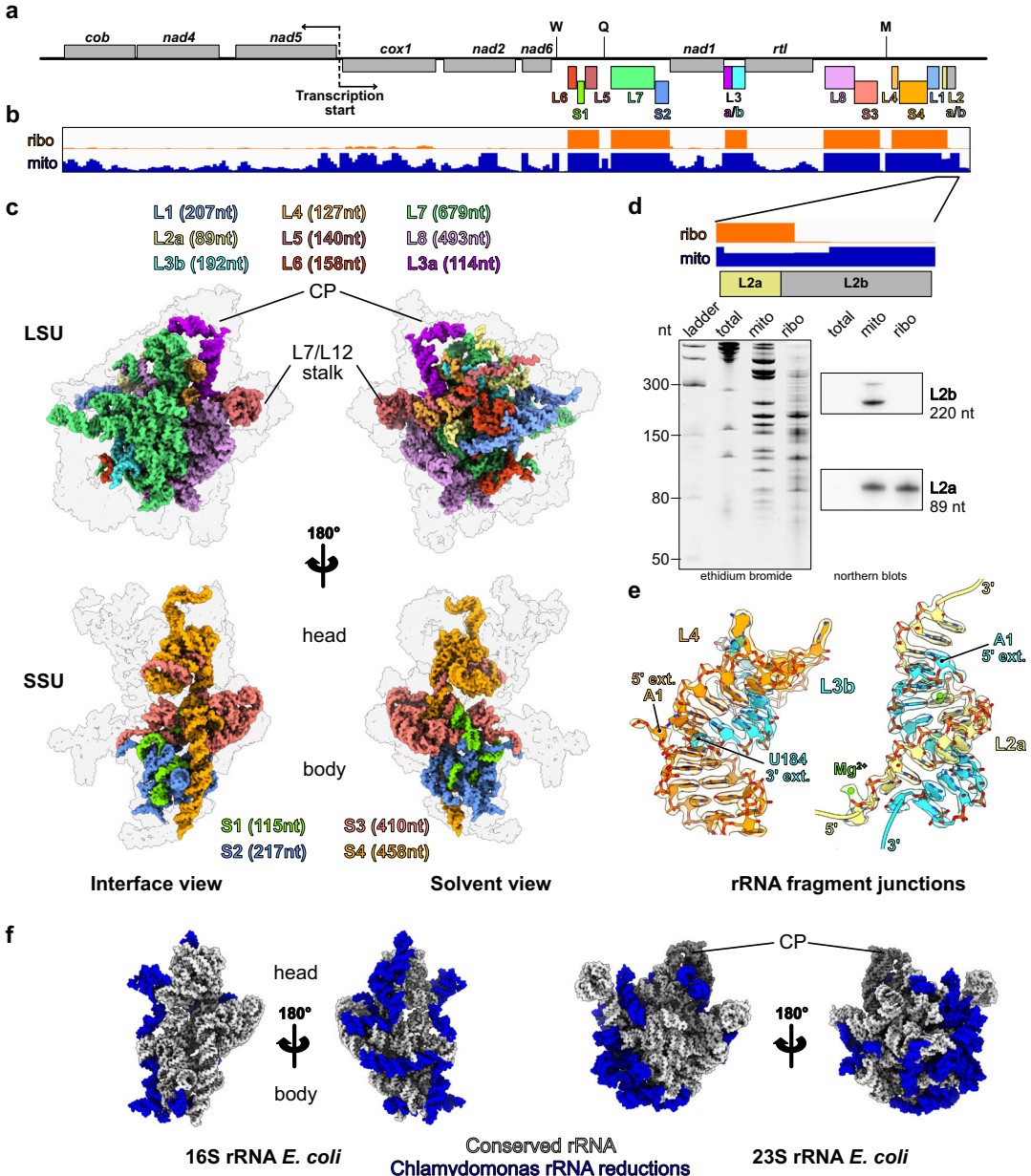

**Fig. 3 The ribosomal RNAs of the *Chlamydomonas* mitoribosome are fragmented. a** Schematic representation of the entire *C. reinhardtii* mitochondrial genome. The protein-coding genes are displayed in gray, and the rRNA fragments incorporated in the mitoribosome are individually colored. The tRNA genes are indicated by letters. **b** Browser view of the RNAseq data of libraries built from purified mitochondria (blue) and purified mitoribosome (orange) fractions. Data range is [0–700], mapped to the *Chlamydomonas* mitochondrial genome in **a**. **c** rRNA fragment 3D organization in the LSU and the SSU. The r-proteins are shown as a gray silhouette. The fragment colors match the color code used in **a**. **d** Zoomed view of the L2a/b coverage (top). The L2b fragment is absent in the ribosomal fraction in the RNAseq analysis, which was confirmed by northern blots (bottom) and the cryo-EM reconstruction. Northern blot was performed for the L2a and L2b fragments on total cell extracts (total), mitochondrial (mito), and mitoribosomal fraction (ribo) on one biological sample. **e** Detailed view of the rRNA extremities of the L3b fragment and its pairing with the L2a and L4 fragments. The atomic models are shown mapped into the cryo-EM densities. **f** Comparison of the *Chlamydomonas* mitoribosome rRNAs with the *E. coli* ribosome. rRNA reductions are shown in blue.

but a consensus of 6 consecutive nucleotides confirmed its origin. Its overall structure is also weakly conserved, with only domain γ retaining its characteristic structure to interact with H38. Domain β is angled differently relative to the domain γ stem, which allows the interaction of the terminal loop of domain β with H87, in contrast to other known ribosome structures. Additionally, domain α could not be fully resolved, but most likely interacts with the putative PPR protein (labeled "PPR*"), possibly stabilizing its 3′ and 5′ termini. In conclusion, when compared with the flowering plant mitoribosome[11], the overall structure of the CP appears similar. However, in terms of composition, the *Chlamydomonas* mitoribosome CP includes a divergent 5S, lacks the uL18m protein, and contains an additional protein, PPR*.

**Specific r-proteins stabilize the rRNAs by highly intertwined protein-RNA interactions.** In the LSU, all the *Chlamydomonas*-specific r-proteins except mL119 are predicted to be nucleic-acid

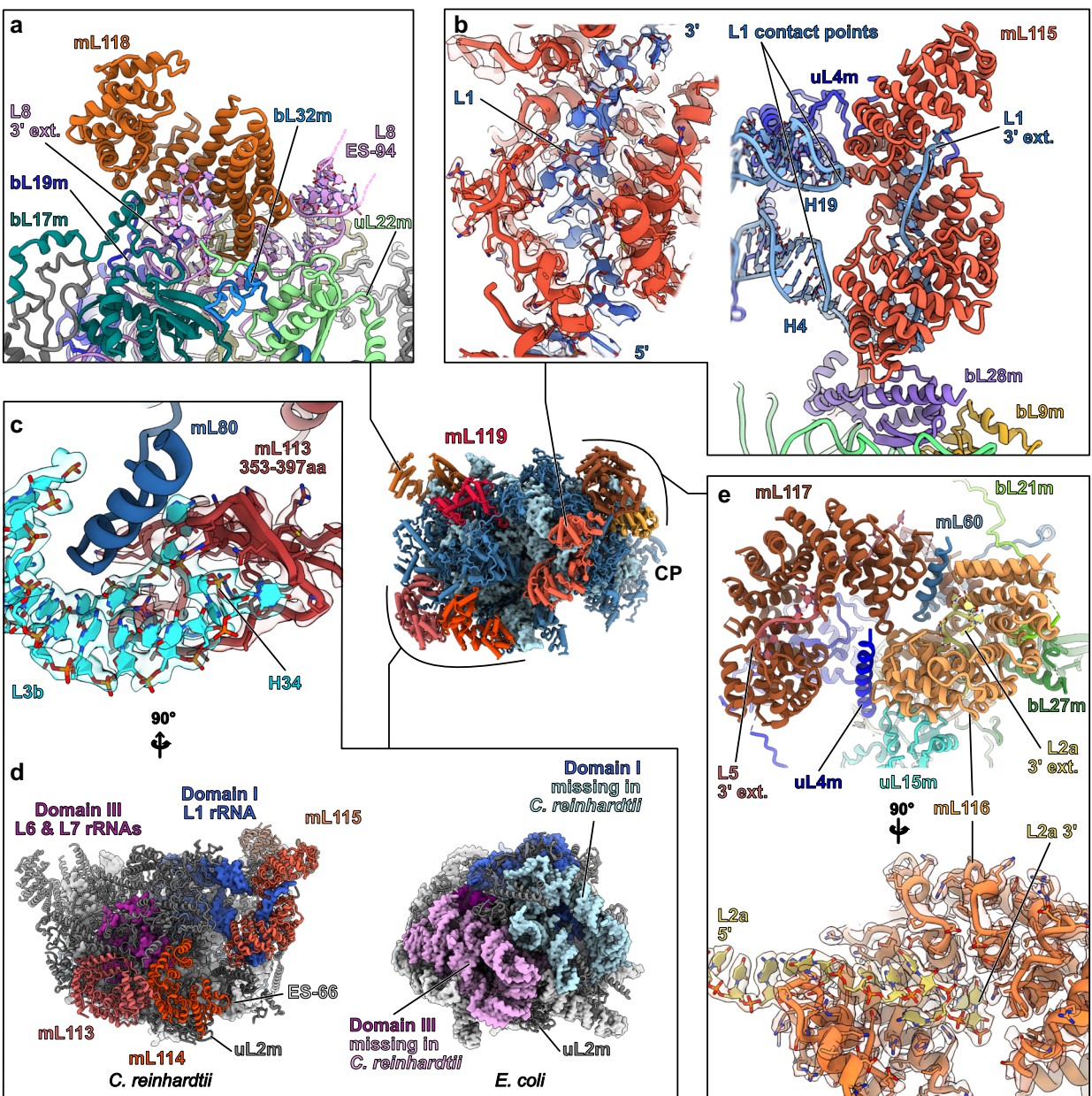

**Fig. 4 *Chlamydomonas*-specific proteins stabilize the fragmented rRNAs via highly intertwined interactions.** Magnified views of the *Chlamydomonas*-specific r-proteins involved in rRNA stabilization. **a** mL118 stabilizes the 3′ end of the L8 fragment and contacts the expansion segment 94 (L8 ES-94). **b** mL115 stabilizes the highly reduced domain I, which is formed by the L1 fragment. mL115 binds the 3′ end of the L1 fragment and also contacts the L1 fragment at two specific points corresponding to H4 and H19. The single-stranded portion of L1 interacting with mL115 is shown in its density. **c** Detailed view of the mL113 contact with the L3b fragment. An inter-repeat domain of mL113 (red) formed by amino acids 353–397 clamps the tip of H34 (cyan). The models are shown in their densities. Contrary to the rest of the r-proteins, mL113 does not enlace single-stranded rRNA. **d** Structural compensation for the loss of large portions of domain I and III. The missing rRNA regions are depicted on the *E. coli* model in pink and light blue, and the compensating proteins mL113, mL114, and mL115 (red shades) are shown on the *Chlamydomonas* model. mL113 and mL114 compensate for domain III reduction, and mL115 stabilizes and compensates for the reduced domain I. **e** mL116 and mL117 interact with each other and with several surrounding proteins. mL117 is involved in stabilizing the 3′ end of the L5 rRNA fragment, and mL116 stabilizes the L2a 3′ extremity, but also makes additional contacts with rRNAs (Supplementary Fig. 4). The single-stranded portion of L2a in mL116 is shown in its density. With the exception of the mTERF protein mL114, the rest of the proteins belong to the same class of ASA2-like/OPR proteins. Further detailed views are shown in Supplementary Fig. 4.

binders. mL114 is an mTERF protein, but mL113, mL115, mL116, mL117, and mL118 all appear to belong to the same protein family, as they all have similar tertiary structures and resemble ASA2/OPR proteins. These OPR proteins (Octo-tricoPeptide Repeat), are predicted to fold into repeated pairs of α-helices, forming a super-helical solenoid, similar to PPR (PentatricoPeptide Repeat) and TPR (TetratricoPeptide Repeat)

proteins[32,33]. Both PPR and TPR are widespread in eukaryotes and have previously been found in mitoribosomes, notably in the flowering plant mitoribosome, which includes 8 PPR proteins (rPPR proteins) that stabilize the numerous rRNA expansions[11]. In *Chlamydomonas*, OPR proteins were previously described to be involved in gene expression regulation, notably in the chloroplast[32,34–37]. In the mitoribosome, these proteins stabilize

the many rRNA fragments by different modes of RNA interaction. This is, to our knowledge, the sole structural description of this kind of protein in interaction with RNA. Proteins mL115, mL116 and mL117 stabilize the 3′ extremities of L1, L2a, and L5, respectively, by binding the single-stranded rRNA fragments in their inner groove (Fig. 4). The stabilization is primarily mediated by positive/negative charge interactions, where the inner grooves of the proteins are largely positively charged, filled with lysine and arginine that interact with the negatively charged phosphate backbone of the RNA (Supplementary Fig. 4). Unlike the rest of the RNA binders, mL113 does not interact with RNA in its inner groove. An inter-repeat domain formed by amino acids 353–397 clamps the tip of H34 from the L3b fragment (Fig. 4c and Supplementary Fig. 4a). Here, the interaction does not stabilize the rRNA itself, but rather constitutes an anchor point between mL113 and the ribosome. mL118 acts similarly to the SSU's OPR mS107, as it binds the 3′ extremity of the L8 fragment, which forms a loop inside the inner groove of the protein (Fig. 4a). Additionally, these proteins also interact with RNA via the convex side of their super-helical fold. This is the case for mL115, which interacts with H4 and H19 of the L1 fragment. mL116 interacts with H28 of the L2a fragment and H38 formed by the L3b/L4 duplex, while mL118 interacts with ES-94 of the L8 fragment (Supplementary Fig. 4). Moreover, the mTERF protein mL114 stabilizes the ES-66 via its C-terminal region, not its inner groove (Supplementary Fig. 4a). All these proteins largely interact with conserved proteins as well as with each other, e.g., mL113 with mL114 and mL116 with mL117 (Fig. 4). Furthermore, mL113, mL114, and mL115 structurally compensate for missing rRNA on the back side of the LSU (Fig. 4d). mL113 and mL114 compensate for the almost wholly deleted domain III, while mL115 both stabilizes and compensates for the missing parts of domain I. Interestingly, mL113 and mL114 are similarly positioned compared to mL101 and mL104 of flowering plants, which stabilize the remodeled domain III[11].

**Knockdown of *Chlamydomonas*-specific r-proteins affects cell fitness and rRNA stability.** Next, we used targeted gene silencing to investigate the importance of the *Chlamydomonas*-specific r-proteins for ribosome integrity. We explored the *Chlamydomonas* CLiP mutant library[38], but no mutant strains for the genes of interest could be confirmed. Hence, we generated artificial miRNA (amiRNA) strains for these factors. This method reduces targeted protein expression at the transcript level[39]. Strains were generated for mL113, mL116, mL117, mL118, and mS105 (p32). The physiological phenotype of each amiRNA strain was analyzed, particularly the capacity to grow under heterotrophic conditions (dark + acetate), which is typically defective in *Chlamydomonas* mutants impaired in mitochondrial respiration[16]. Some transformants revealed growth retardation when cultivated in heterotrophic conditions, and the ones presenting the most severe phenotypes were selected (Fig. 5a). The expression of targeted mRNAs was monitored by quantitative RT-PCR, showing reductions of 80%, 49%, 82%, 51%, and 83% on average for mL113, mL116, mL117, mL118, and mS105, respectively (Fig. 5b).

The levels of rRNA fragments in these downregulated amiRNA strains were then monitored by quantitative RT-PCR to determine the effect on mitochondrial rRNAs stability. This analysis showed that the overall relative levels of LSU rRNAs decreased by 13%, 40%, and 14% in the *mL113, mL117, mL118* knockdown strains, respectively, while the level of L2b RNA, which is not present in mitoribosomes, followed a different behavior (Fig. 5c). In contrast, relative rRNA levels were not significantly affected in the *mL116* and *mS105* knockdown

strains, with the exception of L2b, which was reduced to about 67% in *mS105*. Altogether, it appears that *Chlamydomonas*-specific r-proteins, in particular mL113, mL117, and mL118, are required for the proper stability of the LSU rRNAs. In addition, the accumulation of mitochondria-encoded proteins was investigated by protein immunoblots. Two mitochondrial-encoded components of respiratory complex I, Nad4 and Nad6, were analyzed alongside two controls, the nuclear-encoded subunit NUO7 of complex I and the mitochondrial porin VDAC (Fig. 5d). Analysis from 3 to 4 technical replicates from 2 biological replicates showed that Nad4 and Nad6 levels were decreased in the *mL113, mL117*, and *mS105*-mutant strains compared to wild-type; in contrast, NUO7 and VDAC had unchanged levels (Supplementary Fig. 9 and Source Data file). Finally, the accumulation of assembled respiratory complexes, which contain mitochondria-encoded proteins, was investigated by blue native PAGE (BN-PAGE) coupled to in-gel activity assays (Fig. 5e). These tests revealed that the *mL113* strain is impaired in complex I and IV activity, whereas the *mL117* strain also appears to be affected but to a lesser extent, which correlates with the immunoblot assays. Collectively, these analyses show different impacts on the knockdown strains, suggesting non-redundant functions for these r-proteins.

**The *Chlamydomonas* mitoribosome is tethered to the inner mitochondrial membrane via two protein contact sites.** The LSU reconstruction revealed the presence of a specific r-protein, mL119, precisely located at the exit of the peptide channel (Fig. 6). There, this protein forms several contacts with r-proteins uL22m, uL24m, uL29m, and bL32m, and with nucleotides 114–124 of the L6 fragment via its C-terminal part, which anchors the protein to the ribosome (Fig. 6e). The N-terminal part of mL119, which constitutes most of the protein's mass, is exposed to the solvent. This protein has no apparent homolog and appears to be restricted to the Chlorophyceae (green alga) lineage. Similar to humans, the *Chlamydomonas* mitochondrial genome only codes for membrane components of the respiratory chain, except for the *rtl* gene, whose expression and function remain uncertain. In humans and yeast, it was previously shown that mitoribosomes contact the membrane protein insertase Oxa1 via r-protein mL45 in human and the linker protein Mba1 in yeast[40–44]. These two homologous proteins are positioned at the exit of the peptide channel, where they link the ribosome to the membrane by binding Oxa1, allowing direct insertion of nascent proteins into the membrane. Given the position of mL119, one would expect this protein to fulfill a function similar to mL45 and Mba1. To assess the role of mL119 in membrane binding, *Chlamydomonas* mitoribosomes were directly visualized inside cells using in situ cryo-ET (Fig. 6 and Supplementary Movie 2).

Whole *Chlamydomonas reinhardtii* cells were vitrified, thinned by cryo-focused ion beam (FIB) milling, and imaged by cryo-ET. A representative tomogram depicting a section of a native mitochondrion within a *Chlamydomonas* cell is shown in Fig. 6a–d. ATP synthase dimers, cytosolic ribosomes, and mitoribosomes were automatically localized by template matching, structurally resolved by subtomogram averaging, and then mapped back into the native cellular environment (Fig. 6b). Contrary to cytosolic ribosomes, which crowd the cytoplasm, mitoribosomes have a very low abundance and are localized to the inner mitochondrial membrane. Their low copy number and membrane association highlight the difficulty of purifying these complexes compared to cytosolic ribosomes. Alignment of subtomograms containing mitoribosomes yielded a structure of the native membrane-bound mitoribosome at 31.5 Å resolution (Supplementary Fig. 3). With the exception of dynamic flexible

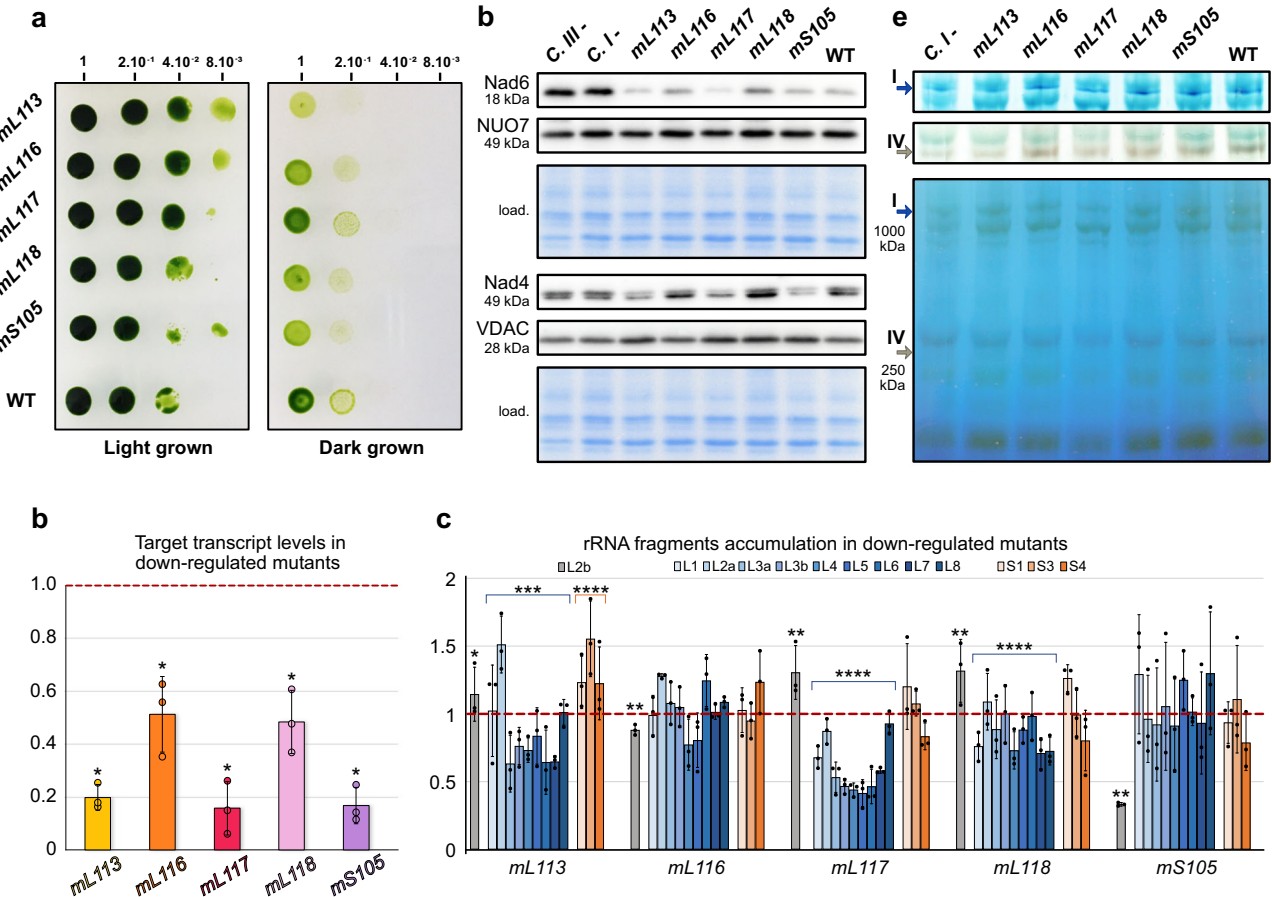

**Fig. 5 Downregulation of *Chlamydomonas*-specific r-proteins affects fitness, rRNA accumulation, and mitochondrial proteins synthesis. a** Growth phenotype of amiRNA knockdown strains. After a first-round selection, transformants were obtained for *mL113, mL116, mL117, mL118,* and *mS105*. Growth phenotypes in the dark were investigated by 5-fold dilution series. Dilutions were spotted on two identical TAP plates, one placed in light and the other in the darkness. **b** Relative levels of amiRNA-targeted mRNAs investigated by RT-qPCR. All strains show clear downregulation compared to WT. Data are presented as mean values ± SD from $n = 3$ biological independent experiments, each analyzed in three technical replicates. The statistical difference between WT and each amiRNA strain for each signal protein was calculated with the two-tailed Mann–Whitney test (*$p < 0.05$). Statistics and the exact $p$-values are detailed in the Source Data file. **c** Relative steady-state levels of rRNA fragments in the different *Chlamydomonas*-specific r-protein amiRNA strains. rRNA fragments of the LSU are shown in blue shades, rRNA fragments of the SSU are shown in orange shades, and L2b (not present in the ribosome) is gray. Data are presented as mean values ± SD from $n = 3$ biological independent experiments, each analyzed in three technical replicates. For statistical analysis, the rRNAs were divided into three groups: the LSU rRNA fragments, the SSU rRNA fragments, and the L2b rRNA fragment. The statistical differences between WT and amiRNA strain for each group was calculated with the two-tailed Mann–Whitney test (****$p < 0.0005$; ***$p < 0.005$; **$p < 0.05$; *$p < 0.5$). Statistics and the exact $p$-values are detailed in the Source Data file. For **b** and **c**, the WT levels are normalized to 1 on the y axis. **d** Steady-state levels of mitochondria-encoded Nad6 and Nad4, as well as nuclear-encoded NUO7 and the mitochondrial porin VDAC, are shown by protein immunoblots corresponding to an example of a typical result from $n = 2$ biological replicates performed in 3–4 technical replicates (Supplementary Fig. 9 and Source Data file). **e** Blue native PAGE and in-gel activity assays for complexes I and IV. Arrows indicate the resulting bands. The activity assay for complex I corresponds to a typical result from two independent experiments. The activity assay for complex IV corresponds to the result of one biological experiment. For **d** and **e**, Wild-type (WT), complex I-mutant *dum5* (C I-), and complex III-mutant *dum11* (C III-) were used as controls.

regions (e.g., L7/L12 and L1 stalks), the in situ subtomogram average is highly similar to our single-particle reconstructions, as revealed by molecular fitting. This structural agreement confirms that the single-particle reconstructions very likely correspond to mature forms of the mitoribosome subunits (Fig. 6f). The in situ subtomogram average had one additional density located at the mitoribosomes's mRNA exit channel. Although we do not know the identity of this density, we speculate that it may correspond to exiting mRNAs or the recruitment of additional factors during active translation (Fig. 6g).

The in situ subtomogram average reveals how the mitoribosome is tethered to the inner mitochondrial membrane. Membrane-bound mitoribosomes were previously described by cryo-ET of mitochondria isolated from yeast[45] and humans[46].

Similar to yeast, but not humans, the *Chlamydomonas* mitoribosome makes two distinct contacts with the membrane. Superposition with the atomic model reveals that one contact is located at the precise position of mL119, supporting the hypothesis that this protein could directly interact with the ribosome binding domain of Oxa1 in vivo. Therefore, it appears that mL119 constitutes a functional analog of mL45 and Mba1. However, mL119 and mL45/Mba1 are not evolutionary related, but rather appear to have convergently evolved to fulfill the same function. The mitoribosome's second membrane contact is mediated via the C-terminal part of mL113 (Fig. 6f). This region of mL113 had poorly resolved density in our cryo-EM map and was thus not modeled, but we could still observe its position at low resolution (Supplementary Fig. 4a). The mL113 contact mimics the rRNA

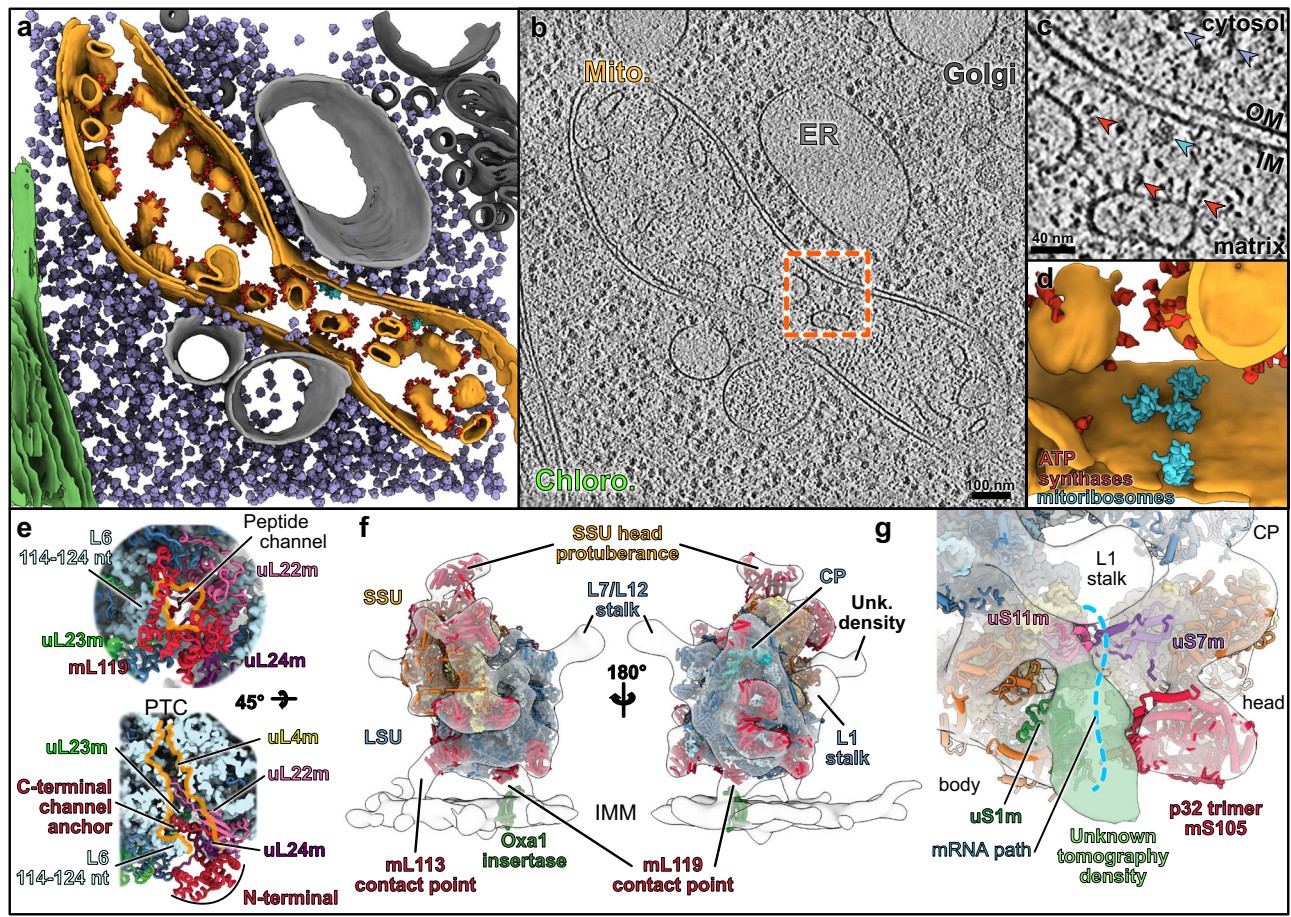

**Fig. 6 The mitoribosome is attached to the mitochondrial inner membrane via specific proteins. a** Segmentation of an in situ cryo-tomogram, depicting a mitochondrion within a native *C. reinhardtii* cell. The mitochondrion is shown in orange, chloroplast in green, ER and peroxisome in light gray, and Golgi in dark gray. Subtomogram averages of ATP synthase dimers (red), cytosolic ribosomes (purple) and mitoribosomes (cyan) are mapped into the volume. **b** A slice through the corresponding raw tomogram (one representative tomogram from n = 47 in total). **c** Close-up view of a membrane-bound mitoribosome at the inner membrane (IM), boxed in **b**. Arrowheads point to ATP synthase, cytosolic ribosomes, and a mitoribosome. **d** Close-up view of the segmented area presented in **c**, highlighting a cluster of three mitoribosomes that possibly form a polysome. Scale bar are indicated on **b** and **c**. **e** The peptide channel of the *Chlamydomonas* LSU from a solvent view (top) and as well as a cut view (bottom). The mL119 protein is located at the exit of the peptide channel. The C-terminal part of mL119 anchors the protein to the ribosome, while the N-terminal part is exposed to the solvent where it could interact with Oxa1. Orange lines delimit the peptide channel. Ribosomal proteins are depicted as cartoons, and rRNAs are depicted as surface representations in light blue. **f** The atomic model of the mitoribosome is fitted into the subtomogram average density of the native membrane-bound mitoribosome. The mitoribosome contacts the inner mitochondrial membrane (IMM) at two specific points. These contacts are mediated by two *Chlamydomonas*-specific proteins, mL113 and mL119, the latter of which is located at the exit of the peptide channel, adjacent to the Oxa1 insertase. The model of the bacterial homolog of Oxa1 is shown for illustration purposes. **g** Close-up view of the mRNA exit channel. The molecular model is fitted into the subtomogram average, highlighting the presence of an additional unknown density (green) located close to uS1m.

expansion segment ES-96 that forms the second contact site with the membrane in yeast[45].

## Discussion

Our study describes the structure and composition of the *Chlamydomonas* mitochondrial ribosome. The cryo-EM reconstructions show that this green alga mitoribosome differs significantly from prokaryotic ribosomes as well as its flowering plant counterpart[47,48]. In both the small and the large subunits, the mitoribosome has acquired several additional r-proteins that significantly reshape its overall architecture. These specific r-proteins combined with the fragmented rRNAs (four pieces in the SSU and nine pieces in the LSU) constitute an extreme case of ribosome divergence, even among the exceptionally diverse mitoribosomes.

One striking feature of the SSU is the presence of two large protuberances on the head and the body. The body protuberance could be assigned to a large insertion in the mitoribosome-specific protein mS45 (Supplementary Fig. 5b), which shows high structural variability between mitoribosomes in different species despite the conservation of its core domain[22]. The head protuberance was poorly resolved in our density map but is most likely composed of long extensions of the head's conserved r-proteins (uS3m, uS10m, mS35). In vascular plant mitoribosomes, the uS3m r-protein was shown to form a similar large protuberance on the SSU head, suggesting a common origin of these protrusions[11]. The roles of these two protuberances are unknown. The position of the body protuberance, close to the mRNA entry channel, might suggest a species-specific mechanism of mRNA recruitment, similar to that mediated by mS39 in humans[4,41,46,49]. This protuberance, in conjunction with the additional density observed in the subtomogram average next to bS1m at the mRNA exit channel (Fig. 6g), might point to the existence of specific translation processes in *Chlamydomonas*.

Translation initiation in *Chlamydomonas* mitochondria shares some features with that of human mitochondria, as the mRNAs lack 5′ untranslated regions in both organisms. However, *Chlamydomonas* most probably has a specific mechanism for translation initiation, as its mitochondrial mRNAs do not have the U-rich motif downstream of the AUG that was proposed to interact with mS39 in humans. Furthermore, mature *Chlamydomonas* mRNAs have poly-C rich 3′ tails that might be required for translation initiation[19].

Another key feature of the SSU is the homotrimeric mS105 (p32) protein forming a torus-shaped protuberance on the back of the SSU head, reminiscent of RACK1 on the cytosolic ribosome[50]. This protein belongs to the MAM33 family, which appears to be eukaryote-specific and is characterized by its quaternary structure: a doughnut-shaped trimer that is highly negatively charged[26,51]. The p32 protein has been the subject of many studies, as its mutations result in severe diseases in humans[25,52–54]. However, despite decades of research, its precise functions remain elusive. Recent studies suggest that MAM33-family proteins might be involved in mitoribosome biogenesis. Indeed, they are linked to LSU biogenesis in yeast[24], the recent structure of the Trypanosoma SSU "assemblosome" includes a heterotrimeric p22 (homolog of p32), directly highlighting its role in mitoribosome biogenesis[55,56], and in humans, the YBEY protein forms a complex with p32 and is involved in SSU biogenesis[23]. We observed that p32 is an integral component of the *Chlamydomonas* mitoribosome. However, it does not appear to have an obvious function related to the translation process, as it does not bind rRNA and only makes a few contacts with the adjacent r-proteins. Likewise, its downregulation does not impair the accumulation of mitochondrial rRNAs. Taking these results together with the above-mentioned studies, it seems that p32 could act in mitoribosome maturation. Given its overall negative charge, it might serve as a binding platform that scaffolds other factors during ribosome biogenesis. It is unclear why p32 would be kept as a constitutive ribosomal protein in *Chlamydomonas* and not in other eukaryotes, but this may point to species-specific functions.

One of the most prominent features of the *Chlamydomonas* mitoribosome is its extensively fragmented rRNAs, with four pieces in the SSU and nine pieces in the LSU. It is interesting to note that Kinetoplastida and Euglenozoa also have fragmented rRNA in their cytosolic ribosomes[57–62]. However, only the LSU rRNA is fragmented, and the fragments are continuous in the genome. Among mitoribosomes, the ciliate mitoribosome also contains LSU and SSU rRNAs that are each split into two pieces, contrasting with the extensive fragmentation observed here with *Chlamydomonas*[63–65].

It has been proposed that the fragmentation and scrambling of the *Chlamydomonas* rRNA genes are the result of several mitochondrial genome recombination events between short repeated sequences[31]. We assigned all the previously identified rRNA fragments in the mitoribosome except one, L2b. This fragment is not incorporated into the mature mitoribosome and is thus not an rRNA. Nevertheless, its transcript has reproducibly been found in the total mitochondrial fraction[19,28]. Its function is unclear, but it might be involved in mitochondrial genome maintenance, which involves telomere-like structures in *Chlamydomonas*, as the L2b sequence is highly similar to both ends of the linear *Chlamydomonas* mitochondrial genome[66,67]. Alternatively, the observation that the L2b RNA level is decreased in the *mS105*-mutant strain may hint at a function related to mS105, and thus, possibly to mitoribosome or SSU biogenesis. We cannot exclude the possibility that the mS105 protein plays a role outside the mitoribosome and is involved in mitochondrial genome maintenance via an interaction with L2b.

Importantly, we reveal that the L3a rRNA fragment is a 5S rRNA, which previously escaped identification because of its highly divergent primary sequence. Even compared to closely related Chlorophytes and Chlorophyceae species, *Chlamydomonas* L3a is particularly different at the sequence level (Supplementary Fig. 8d). L3a occupies the same position as a classical 5S, but its overall structure, notably the domain α and β structures, are quite different from other known 5S structures, rendering it one of the most divergent 5S rRNA described to date. Putative highly divergent mitochondrial 5S rRNAs have also been previously identified in various amoebozoan species[68,69]. Given the high divergence of the 5 S rRNA in *Chlamydomonas*, together with the aforementioned studies, a wider phylogenetic distribution of mitochondrial 5 S rRNA might be suggested.

The rest of the rRNA fragments form the core of the mitoribosome and are globally conserved, yet reduced. In the ribosome core, these fragments are stabilized by base-pairing with each other. In contrast, on the outer shell of the ribosome, especially in the LSU, the rRNA fragments are stabilized by the *Chlamydomonas*-specific r-proteins. These proteins are all alpha-helical repeats belonging to nucleic-acid binder families PPR, OPR, and mTERF. In our structures, they form highly intertwined interfaces with single- and double-stranded RNA, all involving positive/negative charge interactions. These proteins stabilize the 3′ end of rRNA fragments L1, L2a, L5, and L8 by enlacing their single-stranded extremities and also contact and stabilize additional rRNA helices via their convex surfaces.

The function of these proteins was investigated by analyzing downregulation mutants (Fig. 5). All mutants for LSU r-proteins, except the *mL116* mutant, showed reduced levels of LSU rRNA fragments, indicating that these r-proteins are important for the stability of the rRNAs, and thus integrity of the mitoribosome. They seem to play a chaperone-like role, stabilizing and perhaps contributing to the recognition and recruitment of the different rRNA pieces during assembly. In *mS105*, where the protein does not directly interact with rRNAs, the rRNA levels are almost unaffected. Additionally, in downregulation mutants *mL113* and *mL117*, steady-state levels of mitochondria-encoded proteins and active respiratory complexes are decreased, highlighting their importance in translation and impact on mitochondrial metabolism.

Our cryo-ET analysis reveals that *Chlamydomonas* mitoribosomes are bound to the inner mitochondrial membrane. In animals and most eukaryotes, the mitochondrial genome encodes almost exclusively components of the respiratory chain, which are all membrane-embedded proteins. These proteins are co-translationally inserted into the inner mitochondrial membrane to reduce the probability of protein aggregation during transport[70]. To facilitate this process, mitoribosomes are consistently found attached to the inner membrane[45,46]. In mammals, the mitoribosome attachment is mediated by a specific r-protein, mL45, located at the exit of the peptide channel, which links the ribosome to the main insertase of the inner membrane, Oxa1[42,43]. In the yeast *S. cerevisiae*, where one of the mitochondria-encoded proteins is soluble[71], the association is mediated by an mL45 homolog, Mba1, which is not an integral constituent of the mitoribosome, and an expansion segment of H96 directly contacting the membrane[13,40,44]. This is most likely also the case in the yeast *N. crassa*[12]. In *Chlamydomonas*, similarly to mammals, all proteins encoded in the mitochondrial genome are components of the respiratory chain. Therefore, it is not surprising that the *Chlamydomonas* mitoribosome would have acquired a specific r-protein to tether translation to Oxa1. Interestingly, the membrane interaction in *Chlamydomonas* is mediated via two contact points. mL119 forms one contact, and mL113 create a second contact point directly with the membrane,

similar to ES-H96 in yeast. Notably, despite the similar location of the mL113 and ES-H96 contact sites, they are of different molecular nature (protein vs. rRNA) and have been acquired via different evolutionary mechanisms: an expansion of the nuclear genome in case of mL113, compared to the expansion of the mitochondrial gene coding for 23S rRNA in yeast. In light of recent literature suggesting an early expansion of mitoribosomal proteins in eukaryotes[72], this raises the question of whether the second contact site is an isolated case of convergent evolution between green algae and yeast, or whether it is a more universal feature of mitoribosomes that was either replaced (yeast) or lost (mammals) throughout evolution. The fact that membrane association is mediated by different proteins in each organism, yet the Oxa1 contact is conserved, indicates that this interface is particularly critical[40]. In flowering plant mitochondria, which still encode a large number of soluble proteins, accessory factors might recruit mitoribosomes to the membrane, similar to Mba1 in yeast. However, in Tetrahymena, the mitochondrial genome encodes numerous soluble proteins, but the mitoribosome has still acquired a probable permanent anchor to Oxa1, the r-protein mL105[65]. Finally, contrary to mammalian mL45 blocking the peptide channel until mitoribosome association with the membrane[41,43], or kinetoplastid assembly factor mL71 blocking the peptide channel during ribosome maturation[9], the *Chlamydomonas* peptide channel is not blocked by mL119.

In conclusion, our structural and functional characterization of *Chlamydomonas* mitoribosome provides a new perspective on mitoribosome evolution and membrane binding. It delivers essential information to broader questions on the function and evolution of rRNA and the ribosome. This work paves the way for future investigations of mitoribosomes in other species, in particular Apicomplexa such as *Plasmodium* and *Toxoplasma*, where fragmented mitochondrial rRNAs also occur[3,73]. Indeed, the structure of the *Chlamydomonas* mitoribosome demonstrates that despite the extreme fragmentation of rRNAs, the functionally important regions are well preserved. Moreover, it indicates that rRNAs do not have to be covalently continuous if the 3D ribosome structure can be recreated via RNA-RNA and RNA-protein interactions. Interestingly, Gray et al. proposed that long, covalently continuous conventional rRNAs might have evolved from short, non-covalently interacting ancestors[20]. The *Chlamydomonas* mitoribosome might thus represent a relic from an ancestral form of rRNA organization.

Overall, the structure reported here provides further insights into the evolution of mitoribosomes and the elaboration of independent strategies to accomplish and regulate translation. Together with recent work on mitoribsosomes from other species, our study demonstrates how the mitoribosome is truly one of nature's most eclectic playgrounds for evolving diverse strategies to regulate a fundamental cellular process.

## Methods

### *Chlamydomonas reinhardtii* mitochondria and mitoribosome purification.
*Chlamydomonas reinhardtii* cell wall-less strain CC-4351 (cw15−325 arg7−8 mt+) was used for mitochondria purification and transformation. The mitochondrial mutants dum5[74] and dum11[75] were used as a control for the phenotypic growth analysis in the dark, kindly provided by Dr. Remacle (University of Liège) and, respectively, annotated on figures as CI- and CIII-. The strains were grown on Tris-Acetate Phosphate (TAP) solid or liquid medium[14], supplemented with 100 µg/ml of arginine when necessary, under continuous white light (50 µE/m²/s¹), or in the dark. Mitochondria were isolated from liquid cell cultures grown up to the exponential phase[76]. Cells were harvested by centrifugation 10 min $1000 \times g$, and resuspended in 10 ml ice-cold 25 mM phosphate buffer pH 6.5 containing 6% PEG 6000, 0.4% (w/v) bovine serum albumin (BSA), and 0.016% (w/v) digitonin to a final concentration of $3 \times 10^8$ cells/ml. The suspension was warmed rapidly to 30 °C, kept at this temperature for 30 seconds, and cooled to 4 °C. Then the broken cells were pelleted at $2500 \times g$ and washed with 40 ml of ice-cold 20 mM Hepes-KOH pH 7.2 containing 0.15 M mannitol, 2 mM EDTA, 0.1% (w/v) BSA, and 1 mM MgCl₂. After a 2 min $1000 \times g$ centrifugation, the pellet was resuspended in

2 ml of the same solution, stirred vigorously for 45 s, and then 6 ml of 20 mM Hepes-KOH buffer pH 7.2 containing 0.15 M mannitol, 0.8 mM EDTA, and 4 mM MgCl₂ were added. Mitochondria were collected at $12,000 \times g$ for 10 min, resuspended in the same last buffer, and then loaded on a discontinuous Percoll gradient (13%/21%/45%) in MET buffer (280 mM Mannitol, 10 mM Tris-HCl pH 6.8, 0.5 mM EDTA, and 0.1% BSA) and centrifuged for 60 min at $40,000 \times g$. Purified mitochondria were recovered at the 45/21 interface and washed two times in MET buffer by centrifugation at $12,000 \times g$ for 10 min and stored at −80 °C.

Mitoribosome purification was conducted as previously[10,11]. In brief, purified mitochondria were resuspended in Lysis buffer (20 mM HEPES-KOH, pH 7.6, 100 mM KCl, 30 mM MgCl₂, 1 mM DTT, 1.6% Triton X-100, 0.5% n-DDM, supplemented with proteases inhibitors (C0mplete EDTA-free)) to a 1 mg/ml concentration and incubated for 15 min in 4 °C. Lysate was clarified by centrifugation at $25.000 \times g$, 20 min at 4 °C. The supernatant was loaded on a 40% sucrose cushion in Monosome buffer (Lysis buffer without Triton X-100 and 0.1% n-DDM) and centrifuged at $235,000 \times g$, 3 h, 4 °C. The crude ribosomes pellet was resuspended in Monosome buffer and loaded on a 10–30% sucrose gradient in the same buffer and run for 16 h at $65,000 \times g$. Fractions corresponding to mitoribosomes were collected, pelleted, and resuspended in Monosome buffer and analyzed by nanoLC-ESI-MS/MS and cryo-EM (Supplementary Fig. 1).

### Grid preparation.
For the single-particle analyses, 4 µl of the samples at a protein concentration of 1.5 µg/µl was applied onto Quantifoil R2/2 300-mesh holey carbon grid, coated with thin home-made continuous carbon film and glow-discharged (2.5 mA for 20 s). The sample was incubated on the grid for 30 s and then blotted with filter paper for 2 s in a temperature and humidity-controlled Vitrobot Mark IV (T = 4 °C, humidity 100%, blot force 5) followed by vitrification in liquid ethane.

### Cryo-electron microscopy data collection.
The single-particle data collection was performed on a Talos Arctica instrument (ThermoFisher Company) at 200 kV using the SerialEM software for automated data acquisition. Data were collected at a nominal underfocus of −0.5 to −2.5 µm, at magnifications of 36,000× with a pixel size of 1.13 Å for the SSU, and 45,000× with a pixel size of 0.9 Å for the LSU. Micrographs were recorded as movie stacks on a K2 direct electron detector (GATAN Company); each movie stack was fractionated into 65 frames, for a total exposure of 6.5 s corresponding to an electron dose of 45 e−/Å².

### Electron microscopy image processing.
Drift and gain correction and dose weighting were performed using MotionCorr2[77]. A dose-weighted average image of the whole stack was used to determine the contrast transfer function with the software Gctf[78]. The following workflow was processed using RELION 3.0[79]. Initial analyses were performed in CryoSPARC[80] to asses sample composition and to generate ab-initio cryo-EM map. After reference-free 2D classification, for the LSU 346,994 particles were extracted and used for 3D classification into 6 classes (Supplementary Fig. 2). Ab-initio cryo-EM reconstruction generated in CryoS-PARC was low-pass filtered to 30 Å, and used as an initial reference for 3D classification. Two subclass depicting high-resolution features was selected for refinement with 101,291 particles. After Bayesian polishing, the LSU reconstruction reached 3.00 Å resolution. For the SSU reconstruction, a similar workflow was applied. After 2D classification, 445,469 particles were extracted and used for 3D classification into 6 classes. A single subclass depicting high-resolution features was selected for refinement with 40,131 particles. After focus refinement using masks for the head and body of the small subunit, the SSU reconstruction reached a resolution of 4.19 Å for the body and 4.47 Å for the head. Determination of the local resolution of the final density map was performed using ResMap[81].

### Structure building and model refinement.
The atomic model of the *C. reinhardtii* LSU was built into the high-resolution maps using Coot, Phenix, and Chimera. Atomic models from *E. coli* (PDB: 5KCR) and the *A. thaliana* mitoribosome (PDB: 6XYW) were used as starting points for protein identification and modelization, as well as rRNA modelization. The online SWISS-MODEL[82], as well as AlphaFold[27] through the ColabFold service[83], were used to generate initial models for bacterial and mitochondria conserved r-proteins. Models were then rigid body fitted to the density in Chimera[84] and all subsequent modeling was done in Coot[85]. Extensions were built as polyalanine and mutated to the adequate sequences. *Chlamydomonas*-specific proteins for which no model could be generated were first built entirely as polyalanine, then the sequence-from-map Phenix tool[86] was used to identify each of the proteins, and the correct sequences were placed in the densities. For the SSU, due to the lower resolution in comparison to the LSU, all extensions of the homology models were built as polyalanine and unknown densities were built as Ala residues. For refinement, a combination of regularization and real-space refine was performed in Coot for each proteins. The global atomic model was then subjected to real-space refinement cycles using *phenix.real_space_refine* Phenix[86] function, during which protein secondary structures, Ramachandran, and side-chain rotamer restraints were applied. Several rounds of refinement (manual in Coot and automated using the *phenix.real_space_refine*) were performed to obtain the final models, which were validated using the built-in validation tool of

Phenix, based on MolProbity. Refinement and validation statistics are summarized in Supplementary Table 2.

**Cell vitrification and Cryo-FIB milling.** We used *Chlamydomonas reinhardtii* mat3-4 cells (strain CC-3994)[87], which exhibit superior vitrification due to their small size. The strain was acquired from the Chlamydomonas Resource Center, University of Minnesota, St. Paul. Cells were grown until the mid-log phase in Tris-acetate-phosphate (TAP) medium under constant light exposure and bubbling with a normal atmosphere. Vitrification and FIB milling were performed as previously described[88,89]. Using a Vitrobot Mark 4 (FEI), cells in suspension (4 µl of ~1000 cells per µl) were blotted onto R2/1 carbon-coated 200-mesh copper grids (Quantifoil Micro Tools) and plunge frozen in a liquid ethane/propane mixture. Grids were then mounted into Autogrid supports (FEI) and transferred into either a FEI Scios or FEI Quanta dual-beam FIB/SEM instrument. The grids were coated with an organometallic platinum layer by the gas injection system (FEI), and cells were thinned from both sides with a gallium ion beam to a final thickness of ~100–200 nm.

**Cryo-electron tomography data acquisition.** Cellular tomograms were acquired on a 300 kV Titan Krios microscope (FEI), equipped with a post-column energy filter (Quantum, Gatan) and a direct detector camera (K2 summit, Gatan). Tilt series were recorded using SerialEM software[90] with 2° tilt increments from −60° to +60° (in two halves separated at either 0° or −20°), an object pixel size of 3.42 Å, 12 frames per second, a defocus of −4 to −5.5 µm, and a total accumulated dose of ~100 e−/Å.

**Cryo-electron tomography data processing.** Frames from the K2 detector were motion corrected with MotionCor2[77]. Using IMOD software, tilt series were aligned with patch-tracking, and tomograms were reconstructed with weighted back projection. Out of ~130 tomograms, 47 tomograms containing mitochondria were selected. The following workflow is described in Supplementary Fig. 3. An initial structure of the *C. reinhardtii* mitoribosome was obtained by manually picking 103 mitoribosomes from 27 tomograms following template-free alignment by spherical harmonics[91]. The initial map was then used as a template for auto-mated template matching on 47 tomograms with a voxel size of 2.1 nm using PyTOM[92]. To reduce false positives, the highest correlation peaks of the resulting 6-D cross-correlation function localized in the mitochondrial matrix were manu-ally inspected in UCSF Chimera[84], and a set of 222 subvolumes from 27 tomo-grams was obtained. The subvolumes were reconstructed at a voxel size of 6.84 Å, aligned using PyTOM's real-space refinement, and subjected to one round of classification with a mask encompassing the membrane region. This yielded a class of 73 mitoribosomes with a clear membrane density that was subjected to one more round of real-space refinement in PyTOM. For the resulting average, a resolution of 31.6 Å (large ribosomal subunit) and 31.5 Å (small ribosomal subunit) was determined by fourier-shell cross-resolution of the two maps against the maps obtained by single-particle analysis (FSC = 0.33). For the localization of ATP synthases, an initial structure of the *C. reinhardtii* ATP synthase was obtained by manually picking 417 subvolumes from four tomograms and aligning them using spherical harmonics. The obtained map was used as a template for automated template matching on tomograms with a voxel size of 2.1 nm using PyTOM, and the highest correlation peaks were then manually inspected in Chimera to remove false positives. Cytosolic 80S ribosomes were localized in one tomogram by filtering EMDB-1780[93] to 40 Å and using it as a template for template matching on the deconvolved tomogram (tom_deconv; https://github.com/dtegunov/tom_deconv) using PyTOM. Subvolumes were extracted for the 2100 highest correlation peaks, of which 1700 subvolumes were classified as ribosomes by unsupervised, auto-focused 3D classification[94].

**Proteomic analyses of *C. reinhardtii* mitoribosome composition.** Mass spec-trometry analyses of the total, mitochondrial and ribosomal fractions of *C. rein-hardtii* were done at the Strasbourg-Esplanade proteomic platform and performed as previously[10]. In brief, proteins were trypsin digested, mass spectrometry analyses and quantitative proteomics were carried out by nanoLC-ESI-MS/MS analysis on a QExactive+(Thermo) mass spectrometer. Data were searched against the Uni-ProtKB (Swissprot + trEMBL) database restricted to the *C. reinhardtii* taxonomy with a target-decoy strategy (UniProtKB release 2020_03, taxon 3055, 31246 for-ward protein sequences), Proteins were validated respecting FDR < 1% (false dis-covery rate) and quantitative label-free analysis was performed through in-house bioinformatics pipelines.

**Artificial miRNA *C. reinhardtii* strain generation and analyses.** Artificial microRNAs constructs were created according to Molnar et al.[39] as follows: the oligonucleotides were designed using the WMD3 Web MicroRNA Designer soft-ware v3.2 [http://wmd3.weigelworld.org/cgi-bin/webapp.cgi] and genome release *Chlamydomonas CDS reinhardtii* 281 v5.6.cds (Phytozome) (Supplementary Table 3). The oligonucleotides were annealed, phosphorylated, and ligated into a SpeI-digested pChlamiRNA2 containing the ARG7 gene as a selection marker. The resulting plasmids were linearized and transformed into *Chlamydomonas* CC-4351 strain by the Neon® Transfection System (LifeTechnologies) according to the

GeneArt® MAX Efficiency® Transformation protocol for Algae (LifeTechnologies Cat#A24229). Cells with integrated plasmid were selected on TAP plates without arginine. Colonies (16–48 depending on the transformation) were picked to grow to logarithmic phase on liquid TAP medium. They were then spotted on two identical TAP plates to test their capacity to grow in the dark. One plate was placed in a mixotrophic condition (light + acetate) for 5–7 days, and the other one in a heterotrophic condition (dark + acetate), for 10–15 days. For the dilution series, cells were grown for 3–4 days on TAP plates and were resuspended in 2 ml of liquid TAP medium. The cell density was measured spectrophotometrically at OD750 and diluted to an OD750 = 1.5. This normalized suspension was used as the starting material (set to 1) for making three serial 5-fold dilutions (2.10−1, 4.10−2, and 8.10−3). A volume of 10 µl for each dilution was then spotted on two identical TAP plates.

**rRNA analysis by RNA sequencing.** The RNAs were prepared from cells using TRI Reagent® (Molecular Research Center) according to the manufacturer's instructions.

For northern blots, 1 µg of total, mitochondrial and mitoribosome fraction RNA, were separated on 7 M Urea - 8% polyacrylamide gel, transferred onto Amersham Hybond™-N + membrane (GE Healthcare Cat#RPN203B), and hybridized to radiolabelled oligonucleotide probes (Supplementary Table 3) in 6 × SSC, 0.5% SDS at 45 °C. Washing conditions were: 2 times 10 min in 2 × SSC and 1 time 30 min in 2 × SSC, 0.1% SDS at the hybridization temperature. For each specific probe, the signal was detected with the Amersham Typhoon laser scanner (Amersham).

For the quantitative real-time RT-PCR analyses, RNAs were treated with RQ1 RNase-Free DNase (Promega Cat#M6101) according to Promega's protocol, using 0.2 U/µg of RNA. To obtain cDNA, reverse transcription assays were performed according to the manufacturer's instructions with 2,5 µg of total RNA in the presence of 5 µM of oligo(dT) primer (Supplementary Table 3) and 25 ng/µl of Random Primers (Promega Cat#C118A) using the SuperScript™ IV Reverse Transcriptase (Invitrogen Cat#18090010). The RT-qPCR amplification was carried out with the dsDNA-specific dye Takyon™ SYBR® 2X qPCR Mastermix Blue (Eurogentec Cat#UF-FSMT-B0701) and monitored in real-time with a LightCycler 480 instrument (Roche). The primers are listed in (Supplementary Table 3). The delta-delta Ct method was used to calculate the relative RNA abundance with respect to the geometric mean of two RNA references *MAA7* and *CYN19-3*[95].

For the RNA sequencing, the p204 library was built with total mitochondrial RNA. The RNA was first chemically fragmented (4 min) and then enzymatically treated with Antarctic Phosphatase (NEB#M0289S) and T4 Polynucleotide Kinase (NEB#M0201S). Library preparation was done according to the TruSeq Small RNA Sample Preparation Guide #15004197 Rev. F February 2014. The library was sequenced on the Illumina MiSeq sequencer in a paired-end mode of 2 × 75 nt reads. The NGS192-small library was built with the mitoribosome fraction. The RNA was also enzymatically treated with Antarctic Phosphatase and T4 Polynucleotide Kinase. The library was then constructed with the NEBNext multiplex small RNA Library set for Illumina reference E7580 following the manufacturer's instructions. Following PCR amplification, a size selection was performed on a 6% TBE gel to recover the 160–350 bp PCR fragments for sequencing. The NGS192-total library was prepared according to the Truseq Stranded Total RNA with Ribozero Plant kit, starting from the first-strand cDNA synthesis step and omitting the two first depletion and fragmentation steps. The library was sequenced on the Illumina MiSeq sequencer in a paired-end mode of 2 × 110 nt reads. Both libraries were sequenced at the IBMP platform. The reads were mapped to the *Chlamydomonas* mitochondrial genome (EU306622) using Bowtie2 version 2.4.1 with the following options -end-to-end -very-sensitive -N 0 -L 22. Alignments were displayed with the Integrative Genomics Viewer (IGV) with the bigWig format.

**Protein analyses.** The *Chlamydomonas* crude total membrane fractions were obtained according to Remacle et al.[96] as follow: *Chlamydomonas* cells TAP liquid cultures were collected and resuspended to 2–1.6 × 10[8] cells/ml in MET buffer (280 mM mannitol, 0.5 mM EDTA, 10 mM Tris-HCl pH 7) with 1× cOmplete™ Protease Inhibitor Cocktail and then disrupted by sonication (four times 30 s of sonication and 30 s of pause; Bioruptor® Pico, Diagenode). The suspension was centrifuged (10 min at 500 × g, followed by 4 min at 3000 × g) and the protein content of the supernatant was determined by the Bradford method. Equal amounts of protein were separated using 15% SDS-polyacrylamide gel electro-phoresis (PAGE), and transferred to a 0.45 µm PVDF membrane (Immobilon®-P Transfer Membrane; Merck Millipore Cat.#IPVH00010). Specific antibodies were used in immunoblotting and were detected using chemiluminescence (Clarity Western ECL Substrate, Bio-Rad). We used rabbit sera obtained against *Chlamy-domonas reinhardtii* mitochondrial-encoded subunits complex I, Nad4 (1:1000) and Nad6 (1:100), nuclear-encoded subunit complex I, NUO7 (1:2000), and the nuclear-encoded mitochondrial protein VDACI (1:25000). The expected / apparent molecular weight are as follows 49 kDa/50–55 kDa for Nad4, 18 kDa/18 kDa for Nad6, 49 kDa/38 kDa for NUO7, and 28 kDa/28 kDa for VDACI. Blue native polyacrylamide gel electrophoresis (BN-PAGE) analyses were conducted according to Schägger et al.[97] as follow: the crude total membrane fractions were prepared as above with an additional centrifugation at high speed (27,000 × g for 15 min) and

the final pellet was suspended in ACA buffer (375 mM 6-aminohexanoic acid, 25 mM Bis-Tris, pH 7, and, 250 mM EDTA). 0.5 mg of the crude total membrane were first solubilized in the presence of 1,5% (w/v) *n*-dodecyl-β-D-maltoside and then centrifuged for 40 min 14,200 × *g* at 4 °C to remove insoluble matters. 0.65% (w/v) of coomassie serva blue G was then added to the supernatant prior to separation by electrophoresis on a 5% to 12% polyacrylamide gradient BN gel. In-gel detection of Complex I (NADH dehydrogenase) activity was performed using a 100 mM Tris-HCl pH 7.4 buffer containing 200 µM NADH and 0.2% nitro blue tetrazolium (NBT). In-gel detection of Complex IV (cytochrome *c* oxidase) activity was performed using a 10 mM MOPS-KOH pH 7.4 buffer containing 7.5% saccharose, 19 U/ml catalase from bovine liver, 0.1% cytochrome *c,* and 0.01% 3,3′-diaminobenzidine (DAB).

**Figure preparation and data visualization.** Three-dimensional segmentation of ER, mitochondrial, and chloroplast membranes in the cryo-tomogram was performed using EMAN's convolutional neural network for automated annotation[98]. Using the TOM toolbox in matlab[99], the averages of the cytoribosomes, mitoribosomes, and ATP synthases were pasted into the tomogram at the refined coordinates and angles determined by subtomogram analysis. Figures featuring cryo-EM densities as well as atomic models were visualized with UCSF ChimeraX[100] and Chimera[84].

**Statistical information.** Data are presented as mean values ± SD (standard deviation), calculated using Microsoft Excel version 16.43 and GraphPad Prism 8 version 8.4. The *p*-value < 0.05 was considered the threshold for statistical significance. The *p*-value significance intervals (*) are provided within each figure legend, together with the statistical test performed for each experiment: the two-tailed Mann–Whitney. For Fig. 5b, c, derived statistics correspond to the analysis of mean values of *n* = 3 biological replicates. Statistics detailed data (means, standard deviation, *n* values, exact *p*-values) are provided in the Source Data file.

**Reporting summary.** Further information on research design is available in the Nature Research Reporting Summary linked to this article.

## Data availability

The data that support this study are available from the corresponding authors upon reasonable request. The cryo-EM maps of *C. reinhardtii* mitoribosome have been deposited at the Electron Microscopy Data Bank (EMDB): EMD-13480 for the LSU, EMD-13481 for the head of the SSU, EMD-13477 for the body of the SSU, and EMD-13578 for the subtomogram averaging of the whole ribosome. The corresponding atomic models have been deposited in the Protein Data Bank (PDB) under the accession 7PKT for the LSU and 7PKQ for the SSU. Mass spectrometric data have been deposited to the ProteomeXchange Consortium via the PRIDE partner repository with the dataset identifier PXD024708. RNAseq data were deposited in the NCBI Gene Expression Omnibus under accession number GSE171125. Source data are provided with this paper.

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

## Acknowledgements

We are grateful to Pr. Claire Remacle (University of Liège) for her kind gifts of strains and antibodies. We also thank, J. Chicher and P. Hamman of the Strasbourg Esplanade proteomic platform for the proteomic analysis, and S. Graindorge and D. Pflieger of the IBMP bioinformatics core facility for the bioinformatic analysis. We thank Miroslava Schaffer and Wojciech Wietrzynski for help with FIB milling and cryo-ET acquisition. We thank Jürgen Plitzko and Wolfgang Baumeister for access to cryo-EM instrumentation and support. This work has benefitted from the facilities and expertise of the Biophysical and Structural Chemistry platform (BPCS) at IECB, CNRS UMS3033, Inserm US001, University of Bordeaux. The mass spectrometry instrumentation was funded by the University of Strasbourg, IdEx "Equipement mi-lourd" 2015. This work was supported by a European Research Council Starting Grant (TransTryp ID:759120) to Y.H., by the LabEx consortium 'MitoCross' (ANR-11-LABX-0057_MITOCROSS), by the ITI 2021-2028 program of the University of Strasbourg, CNRS, and Inserm supported by IdEx Unistra (ANR-10-IDEX-0002), and EUR IMCBio (ANR-17-EURE-0023) under the framework of the French Investments for the Future Program to P.G. and L.D., and by the Agence Nationale de la Recherche (ANR) grants [MITRA, ANR-16-CE11-0024-02], [DAMIA, ANR-20-CE11-0021], and [ARAMIS, ANR-21-CE12] to Y.H., P.G., and L.D. Additional funding was provided by the Max Planck Society and the Helmholtz Zentrum München to F.W. and B.E., and the Nederlandse Organisatie voor Wetenschappelijke Onderzoek (Vici 724.016.001) to F.F., as well as by an Alexander von Humboldt Postdoctoral Fellowship to F.W.

## Author contributions

F.W., L.D., P.G., and Y.H. designed and coordinated the experiments. H.M. and T.S.-G. generated the amiRNA strains and analyzed them. T.S.-G. purified mitochondria. F.W. purified the mitochondrial ribosomes. H.S. and F.W. acquired the cryo-EM data and performed the single-particle analysis. R.E., S.P., F.F., and B.E. collected and analyzed the cryo-ET data, including subtomogram averaging. F.W. built the atomic models and interpreted the structures. L.K. performed the mass spectrometry experiments. F.W. prepared and assembled the figures. F.W. wrote the manuscript, which was edited by T.S.-G., R.E., H.S., H.M., L.K., S.P., F.F., B.E., P.G., L.D., and Y.H.

## Competing interests

The authors declare no competing interests.
