## [Peer Review File · Nature Communications]

The Chlamydomonas mitochondrial ribosome: how to build a ribosome from RNA fragmentsReviewers' Comments:

Reviewer #1:

Remarks to the Author:

The authors describe the 3-D structure (determined by cryo-EM) of the mitochondrial ribosome (mitoribosome) of the chlorophyte alga, *Chlamydomonas reinhardtii*. This mitoribosome is notable in that it contains large subunit (LSU) and small subunit (SSU) rRNAs that are not covalently continuous, as in conventional ribosomes, but instead consist of fragments whose coding modules are scrambled together with protein-coding and tRNA genes in the *C. reinhardtii* mitochondrial genome. How these fragments are assembled into a functional mitoribosome, and what this structure looks like, have been outstanding questions that have awaited the development of methodologies such as cryo-EM. What the authors present here is, in my opinion, a tour de force, and a fitting denouement for a discovery that was initially reported more than three decades ago.

I am not a structural biologist and so I am not able to comment critically on the technical aspects of obtaining the cryo-EM structure reported here. Instead, I will confine my remarks to the system itself and its significance, from an historical perspective, with regard to ribosome function and evolution.

Specific Comments:

1. At the end of the 2nd para, of the Introduction, the authors state that "it is enigmatic how these fragments are recruited, interact with each other, and are stabilized to form the 3D mitoribosome structure". This statement underplays predictions made in the original description of this novel gene arrangement (Boer and Gray, 1988) in which detailed SSU and LSU rRNA 2ary structures were presented showing how the separate rRNA fragments could interact with one another via long-range pairing to recapitulate the underlying structure of a conventional covalently continuous rRNA. Indeed, the authors report that the fragmented SSU rRNAs are "mainly stabilized by base pairing with each other", and 2ary structures presented in Figs. S5 and S6 basically parallel the predicted pairings contained in the 1988 publication. The *Chlamydomonas* system was one of the first to demonstrate that an rRNA need not be covalently continuous in order to be functional, while also suggesting a model of rRNA evolution from small RNA pieces (for a recent review, see Gray MW, Gopalan V. Piece by piece: Building a ribozyme. *J Biol Chem.* 2020;295(8):2313-23.). These points deserve more emphasis in this paper.

2. Final sentence of the Introduction: "Our study provides the first example of a ribosome composed of numerous rRNA fragments, revealing a strikingly divergent blueprint for building this conserved molecular machine." Although this is the first example of a MITORibosome composed of numerous rRNA fragments, several cryo-EM structures of cytosolic ribosomes containing fragmented rRNAs have recently been published, including one cited by the authors (Matzov et al., 2020). As well, the ciliate mitoribosome, also cited by the authors (Tobiasson and Amunts, 2020), contains LSU and SSU rRNAs that are each split into two pieces (Schnare MN, Heinonen TY, Young PG, Gray MW. A discontinuous small subunit ribosomal RNA in *Tetrahymena pyriformis* mitochondria. *J Biol Chem.* 1986;261(11):5187-93; Heinonen TYK, Schnare MN, Young PG, Gray MW. Rearranged coding segments, separated by a transfer RNA gene, specify the two parts of a discontinuous large subunit ribosomal RNA in *Tetrahymena pyriformis* mitochondria. *J Biol Chem.* 1987;262(6):2879-87).

3. The identification of RNA fragment L3a as a highly divergent 5S rRNA is a notable finding: kudos to the authors for making this connection! Putative highly divergent mitochondrial 5S rRNAs, not readily recognized as such, have also been identified in various amoebozoan species (Bullerwell CE, Schnare MN, Gray MW. Discovery and characterization of *Acanthamoeba castellanii* mitochondrial 5S rRNA. *RNA.* 2003;9(3):287-92; Bullerwell CE, Burger G, Gott JM, Kourennaia O, Schnare MN, Gray MW. Abundant 5S rRNA-like transcripts encoded by the mitochondrial genome in Amoebozoa. *Eukaryot Cell.* 2010;9(5):762-73). These findings suggest a wider phylogenetic distribution of mitochondrial 5S rRNA than has been appreciated up to now, a point the authors may wish to emphasize.

4. In this same section, I'm puzzled by the wording of the sentence, "Compared to flowering plants (Waltz et al., 2020a), the Chlamydomonas CP is similar in structure but has retained a canonical bacterial 5S rRNA." In fact, the plant mitoribosome contains what is arguably a 'canonical bacterial 5S rRNA', whereas the 5S rRNA retained by the Chlamydomonas CP is, as demonstrated here, highly divergent. The authors should re-word this sentence to clarify what they really mean to say.

5. Were the authors able to obtain any information about modified nucleosides in the Chlamydomonas mitochondrial rRNAs through their cryo-EM analysis? As far as I'm aware, no data in that regard are currently available. Particularly interesting would be the presence/absence of the highly conserved tandem N6,N6-dimethyladenosine residues in the loop of the final helix at the 3'-end of the SSU rRNA.

6.. Final para. of Results: change "convergently evolution" to "convergently evolved"

7. Para. 4 of Discussion: The initial characterization of the *Euglena gracilis* fragmented cytosolic LSU rRNA was Schnare MN, Gray MW. Sixteen discrete RNA components in the cytoplasmic ribosome of *Euglena gracilis*. *J Mol Biol.* 1990;215(1):73-83. This paper should be cited instead of or in addition to the Greenwood and Gray (1998) one. The fragmented cytosolic LSU rRNA in the kinetoplastid *Crithidia fasciculata* was characterized in Gray MW. Unusual pattern of ribonucleic acid components in the ribosome of *Crithidia fasciculata*, a trypanosomatid protozoan. *Mol Cell Biol.* 1981;1(4):347-57, and its gene organization in Spencer DF, Collings JC, Schnare MN, Gray MW. Multiple spacer sequences in the nuclear large subunit ribosomal RNA gene of *Crithidia fasciculata*. *EMBO J.* 1987;6(4):1063-71.

8. The authors might consider combining the Results and Discussion sections as a way of eliminating the considerable information overlap that now exists between the two sections. This would tighten the presentation and allow for the inclusion of a final brief Conclusions section that could be used to emphasize the particular insights that this fragmented system brings to the broader issues of rRNA/ribosome function and evolution (e.g., rRNAs need not be covalently continuous as long as 3D ribosome structure can be recapitulated by RNA-RNA and RNA-protein interactions; functionally critical regions – e.g., peptidyltransferase center – are highly conserved in spite of fragmentation; conventional long covalently continuous rRNAs may have evolved from small non-covalently-interacting ancestors).

Reviewer #2:

Remarks to the Author:

Overview:

This manuscript uses a combination of biophysical and proteomics approaches to reveal the structural composition and organization of the *Chlamydomonas reinhardtii* mitochondrial ribosome (mitoribosome). Specifically, the authors use single particle cryo-electron microscopy to solve the structure of the mitoribosome at resolution sufficient to build an atomic model. They complement structural analyses with mass spectrometry to confirm associations between all ribosomal RNA and protein subunits. Their model shows how several novel proteins associate with and stabilize fragmented components of ribosomal RNAs (rRNA) within the mitoribosome. They further characterize the importance of these interactions using targeted amiRNA silencing. Using cellular cryo-electron tomography, the authors reveal the sub-organellar localization of mitoribosomes, demonstrating that the majority associate with the mitochondrial inner membrane (IMM) through *Chlamydomonas*-specific tethering proteins.

Strengths and significance:

One of the major successes of the paper is the use of complementary experimental strategies to both elucidate and validate the associations between all mitoribosome components. This study exemplifies the type of combinatorial analysis strategy that takes full advantage of each technique, resulting in a

comprehensive and thorough analysis of mitoribosome structure, contextualized by sub-organellar localization. With each experimental system, the authors test the importance of specific interactions, and shed light on how all subunits come together to form the fully-functional mitoribosome. The structure shows several distinct features from previously-described mitoribosome composition, thus suggesting a new perspective on mitoribosome evolution and binding. The discovery that *Chlamydomonas*-specific proteins facilitate attachment of mitoribosomes to the IMM opens up exciting possibilities to explore the functional implications of this tethering, and also potentially the mechanisms mediating mitoribosome assembly in distinct cellular conditions (e.g. changing environments, cellular stress, etc.). I anticipate this paper will impact the mitochondrial and ribosomal biology fields by setting the stage for future investigation of mitoribosome assembly, both within *Chlamydomonas* and in other diverse systems. Furthermore, I anticipate the combinatorial cryo-EM/cellular cryo-ET/MS approach will help guide the development of similar multi-faceted experimental strategies to study a wide range complex macromolecular machines, both at high resolution and within their native cellular environment.

Suggestions for improvement:

Overall, the manuscript is well-written, thorough, and comprehensive, and does not require any major revisions for publication. However, there are several minor points that should be addressed to improve the data visualization and interpretability of the manuscript:

The 8 novel proteins mentioned in the text are difficult to locate in the corresponding Figures 1, 2, S3 and Table S1. There appear to be 11 proteins in Table S1 that are specific to *C. reinhardtii*, and it would be helpful to annotate/specify which of these are 8 novel proteins in the text and by highlighting/denoting these proteins in the figures.

One interesting, unaddressed observation in Figure 5A is that some of the amiRNA knockdown strains appear to have more robust growth in light conditions relative to WT. Could the authors comment on the relevance of this observation? Does it have any functional implications?

For Figure 5, the authors appropriately use RT-qPCR to measure the relative levels of mRNA, however there is a notable lack of assessment to measure efficiency of knockdown at the protein level. It would be important to assess relative levels of relevant proteins in the amiRNA knockdown strains.

The details provided in Figure S2 regarding the single-particle data processing workflow are extremely informative and will help guide future studies. A similar workflow figure outlining the steps for subtomogram averaging and contextual structure-mapping (Figure 6) would improve the overall clarity and reproducibility of these results. This is especially pertinent since many steps in the subtomogram averaging workflow are not as standardized as single-particle analysis, and there is growing interest among users to apply these advanced data processing techniques to study protein structures in cells.

The dashed lines indicating portions that could not be modeled in Figure S5 are difficult to see, perhaps it would be better to indicate with colored boxes.

Figure S1 could be improved by adding in a legend or key to define coloring used in heatmap. We also encourage the authors to consider changing colors to make it more accessible to readers with color-blindness.

Reviewer #3:

Remarks to the Author:

Waltz and colleagues present the cryo-EM structures of the small and the large mitochondrial

ribosomal subunits from the unicellular green algae *Chlamydomonas reinhardtii*, a well-studied model organism with fragmented mitochondrial ribosomal RNAs. The authors identified novel, *Chlamydomonas*-specific, ribosomal proteins, belonging to a variety of RNA binder protein families, that either stabilize the rRNA fragments or structurally compensate for the RNA loss, resulting in a significantly altered overall ribosome architecture. The amiRNA silencing experiments confirmed the role of these proteins for the maintenance of ribosomal structure integrity. In addition, the authors revealed that mitochondrial ribosomes are tethered to the inner mitochondrial membrane through *Chlamydomonas*-specific ribosomal proteins, further expanding our knowledge of mitochondrial translation machinery.

This well-written manuscript is accompanied with clear figures and is therefore suitable for publication upon addressing the following issues:

- table S2 statistics are mostly reasonable, however, the percent of the allowed Ramachandran regions is somewhat high (15 %) - aim should be below 5%. Please, add also RNA validation score for Pucker, bond, angle and suite outliers. The authors should also provide the reviewers maps and validated coordinates once the revised manuscript is submitted.
- although the authors thoroughly present the overall architectural features of this structurally divergent ribosome, it is unclear how the fragmented ribosomal RNA and perhaps newly recruited ribosomal proteins affect the functional sites, such as the decoding center and the peptidyl transferase center. This should be explicitly stated either in the results' Overall structure section or in the Discussion.
- the authors should double check if the correct SSU maps have been used for calculating the map to model FSC in figure S2. The sudden drop in the plots may indicate the use of filtered maps.
- the last two sentences of the second paragraph in the knock-down results chapter with references to figures 5A and 5B are duplicated.
- the authors established that the L2b RNA fragment is not a part of ribosomal RNA and is linked to mitochondrial genome maintenance, however, they should then elaborate more why the mS105 knockdown significantly lowers the relative L2b amounts. figures 6E and 6F should be larger.

Response to reviewers' comments:

Reviewer #1

We thank reviewer 1 for the positive appreciation of our work.

Specific Comments:

1. At the end of the 2nd para, of the Introduction, the authors state that “it is enigmatic how these fragments are recruited, interact with each other, and are stabilized to form the 3D mitoribosome structure”. This statement underplays predictions made in the original description of this novel gene arrangement (Boer and Gray, 1988) in which detailed SSU and LSU rRNA 2ary structures were presented showing how the separate rRNA fragments could interact with one another via long-range pairing to recapitulate the underlying structure of a conventional covalently continuous rRNA. Indeed, the authors report that the fragmented SSU rRNAs are “mainly stabilized by base pairing with each other”, and 2ary structures presented in Figs. S5 and S6 basically parallel the predicted pairings contained in the 1988 publication. The *Chlamydomonas* system was one of the first to demonstrate that an rRNA need not be covalently continuous in order to be functional, while also suggesting a model of rRNA evolution from small RNA pieces (for a recent review, see Gray MW, Gopalan V. Piece by piece: Building a ribozyme. *J Biol Chem.* 2020;295(8):2313-23.). These points deserve more emphasis in this paper.

We thank the reviewer for this comment. The predictions made by the authors where indeed remarkably accurate and in line with what is observed in our study. This is now clearly stated in the main text introduction (lines 69-70).

2. Final sentence of the Introduction: “Our study provides the first example of a ribosome composed of numerous rRNA fragments, revealing a strikingly divergent blueprint for building this conserved molecular machine.” Although this is the first example of a MITORibosome composed of numerous rRNA fragments, several cryo-EM structures of cytosolic ribosomes containing fragmented rRNAs have recently been published, including one cited by the authors (Matzov et al., 2020). As well, the ciliate mitoribosome, also cited by the authors (Tobiasson and Amunts, 2020), contains LSU and SSU rRNAs that are each split into two pieces (Schnare MN, Heinonen TY, Young PG, Gray MW. A discontinuous small subunit ribosomal RNA in *Tetrahymena pyriformis* mitochondria. *J Biol Chem.* 1986;261(11):5187-93; Heinonen TYK, Schnare MN, Young PG, Gray MW. Rearranged coding segments, separated by a transfer RNA gene, specify the two parts of a discontinuous large subunit ribosomal RNA in *Tetrahymena pyriformis* mitochondria. *J Biol Chem.* 1987;262(6):2879-87).

This is indeed true. Citations were concerning the initial identification of the fragmented rRNAs in ciliate and the sentence “Among mitoribosomes, the ciliate mitoribosome also contains LSU and SSU rRNAs that are each split into two pieces, contrasting with the extensive fragmentation observed here with *Chlamydomonas*” was added in the discussion (lines 374-376).

3. The identification of RNA fragment L3a as a highly divergent 5S rRNA is a notable finding: kudos to the authors for making this connection! Putative highly divergent mitochondrial 5S rRNAs, not readily recognized as such, have also been identified in various amoebozoan species (Bullerwell CE, Schnare MN, Gray MW. Discovery and characterization of *Acanthamoeba castellanii* mitochondrial 5S rRNA. *RNA.* 2003;9(3):287-92; Bullerwell CE, Burger G, Gott JM, Kourennaia O, Schnare MN, Gray MW. Abundant 5S rRNA-like transcripts encoded by the mitochondrial genome in Amoebozoa. *Eukaryot Cell.*

2010;9(5):762-73). These findings suggest a wider phylogenetic distribution of mitochondrial 5S rRNA than has been appreciated up to now, a point the authors may wish to emphasize.

We appreciate the reviewer comment. We now emphasize that part in the Discussion (lines 394-397).

4. In this same section, I'm puzzled by the wording of the sentence, "Compared to flowering plants (Waltz et al., 2020a), the *Chlamydomonas* CP is similar in structure but has retained a canonical bacterial 5S rRNA." In fact, the plant mitoribosome contains what is arguably a 'canonical bacterial 5S rRNA', whereas the 5S rRNA retained by the *Chlamydomonas* CP is, as demonstrated here, highly divergent. The authors should re-word this sentence to clarify what they really mean to say.

The sentence was indeed misleading and is now reworded.

5. Were the authors able to obtain any information about modified nucleosides in the *Chlamydomonas* mitochondrial rRNAs through their cryo-EM analysis? As far as I'm aware, no data in that regard are currently available. Particularly interesting would be the presence/absence of the highly conserved tandem N6,N6-dimethyladenosine residues in the loop of the final helix at the 3'-end of the SSU rRNA.

This information would indeed be of great interest for the community. Currently, these modifications and their roles are best described in human. One nice recent study would be Pedro Rebelo-Guimar, *et al*, A late-stage assembly checkpoint of the human mitochondrial ribosome large subunit, currently deposited as a preprint on BioRxiv, where they describe the role of 2'-O-methylation in the maturation of the human large subunit. However, seeing these modifications require higher resolution to be sure of the interpretation (2.6 Å and above) than what we managed to obtain here.

6. Final para. of Results: change "convergently evolution" to "convergently evolved"

This has been corrected.

7. Para. 4 of Discussion: The initial characterization of the *Euglena gracilis* fragmented cytosolic LSU rRNA was Schnare MN, Gray MW. Sixteen discrete RNA components in the cytoplasmic ribosome of *Euglena gracilis*. *J Mol Biol.* 1990;215(1):73-83. This paper should be cited instead of or in addition to the Greenwood and Gray (1998) one. The fragmented cytosolic LSU rRNA in the kinetoplastid *Crithidia fasciculata* was characterized in Gray MW. Unusual pattern of ribonucleic acid components in the ribosome of *Crithidia fasciculata*, a trypanosomatid protozoan. *Mol Cell Biol.* 1981;1(4):347-57, and its gene organization in Spencer DF, Collings JC, Schnare MN, Gray MW. Multiple spacer sequences in the nuclear large subunit ribosomal RNA gene of *Crithidia fasciculata*. *EMBO J.* 1987;6(4):1063-71.

The correct citations were added.

8. The authors might consider combining the Results and Discussion sections as a way of eliminating the considerable information overlap that now exists between the two sections. This would tighten the presentation and allow for the inclusion of a final brief Conclusions section that could be used to emphasize the particular insights that this fragmented system brings to the broader issues of rRNA/ribosome function and evolution (e.g., rRNAs need not be covalently continuous as long as 3D ribosome structure can be recapitulated by RNA-RNA and RNA-protein interactions; functionally critical regions – e.g., peptidyltransferase center – are highly conserved in spite of fragmentation; conventional long covalently continuous rRNAs may have evolved from small non-covalently-interacting ancestors).

We agree with the reviewer that a final brief conclusion underlining the insights that the fragmented *Chlamydomonas* system contributes to the ribosome field would be beneficial.

Therefore, we expanded the final conclusion (lines 448-463) to highlight this aspect. However, respectfully and after concertation amongst the authors, we would like to maintain the current organization of the article by first describing the structure of mitoribosomes in the results and then highlighting and discussing some unique features. We feel that our current organization makes the manuscript more intelligible to a broader readership.

Reviewer #2 :

We thank reviewer 2 for their positive appreciation of our work.

Suggestions for improvement:

Overall, the manuscript is well-written, thorough, and comprehensive, and does not require any major revisions for publication. However, there are several minor points that should be addressed to improve the data visualization and interpretability of the manuscript:

The 8 novel proteins mentioned in the text are difficult to locate in the corresponding Figures 1, 2, S3 and Table S1. There appear to be 11 proteins in Table S1 that are specific to *C. reinhardtii*, and it would be helpful to annotate/specify which of these are 8 novel proteins in the text and by highlighting/denoting these proteins in the figures.

The novel proteins identified in the *Chlamydomonas* mitoribosome, 8 in the LSU and 3 in the SSU, are now clearly listed/named in the text and are now highlighted in Figure 2 with underlined names.

One interesting, unaddressed observation in Figure 5A is that some of the amiRNA knockdown strains appear to have more robust growth in light conditions relative to WT. Could the authors comment on the relevance of this observation? Does it have any functional implications?

In our view, the most important observations relate to the mutant's growth phenotypes in the dark because they can reveal mitochondrial respiration defects. The growth in the light is mainly shown here as a control to validate the dark growth phenotypes. Still, we agree with the reviewer that the light growth conditions that we show in Fig 5A for some amiRNA knockdown strains have a more robust growth phenotype than wild-type. This phenotype might relate to mitochondrial dysfunction that would have resulted in indirect effects, possibly at the level of chloroplasts, to result in enhanced light growth. Long-term research on the functional relation between mitochondrial translation and other cellular processes will be expected to give clues to explain our observations.

For Figure 5, the authors appropriately use RT-qPCR to measure the relative levels of mRNA, however there is a notable lack of assessment to measure efficiency of knockdown at the protein level. It would be important to assess relative levels of relevant proteins in the amiRNA knockdown strains.

As the reviewer indicates, it would indeed be informative to assess the different amiRNA knockdown targets at the protein level, in addition to their assessment at the mRNA level, as shown in Fig. 5. At this stage, we could not perform this analysis since we have not yet generated specific antibodies corresponding to mL113, mL116, mL117, mL118, and mS105. The obtention of these antibodies and their use for in-depth functional analyses of the respective proteins are the subject of ongoing long-term research in our teams.

Nonetheless, we have now revised the manuscript by providing quantifications at the protein level for Nad6, Nad4, NUO7, and VDAC that were measured by immunoblotting experiments in mL113, mL116, mL117, mL118, mS105, and WT strains. The new quantifications, as well as a statistical analysis performed with 3-4 technical replicates from 2 biological replicates,

showing the statistically significant decreases of Nad4 and Nad6 in mutant strains, are now provided as a new Fig. S9 and mentioned in the text on lines 272-275.

Beyond the quantifications at the protein level, we now also provide new statistical analyses of the RT-qPCR data shown in Fig. 5 for both the amiRNA target mRNAs and the rRNA fragments in the respective mutant strains. This shows that LSU rRNA fragments are indeed statistically significantly decreased in mL113, mL117, and mL118 in contrast with SSU rRNAs that do not show any statistically significant variations in these strains. Statistical analyses are now mentioned in the legend to Fig. 5 (lines 518-522).

The details provided in Figure S2 regarding the single-particle data processing workflow are extremely informative and will help guide future studies. A similar workflow figure outlining the steps for subtomogram averaging and contextual structure-mapping (Figure 6) would improve the overall clarity and reproducibility of these results. This is especially pertinent since many steps in the subtomogram averaging workflow are not as standardized as single-particle analysis, and there is growing interest among users to apply these advanced data processing techniques to study protein structures in cells.

We have now included an additional Figure, Figure S3, which present a visual summary of the subtomogram averaging workflow, similar to Figure S2.

Figure S1 could be improved by adding in a legend or key to define coloring used in heatmap. We also encourage the authors to consider changing colors to make it more accessible to readers with color-blindness.

Figure S1 was revised to include a color scale and the colors were changed to be more accessible to readers with color-blindness.

Reviewer #3 (Remarks to the Author):

We thank reviewer 3 for their positive appreciation of our work.

- table S2 statistics are mostly reasonable, however, the percent of the allowed Ramachandran regions is somewhat high (15 %) - aim should be below 5%. Please, add also RNA validation score for Pucker, bond, angle and suite outliers. The authors should also provide the reviewers maps and validated coordinates once the revised manuscript is submitted.

The models were improved and the Table S2 revised. The models and maps were deposited on the PDB/EMDB under the following accessions EMD-13480 for the LSU, EMD-13481 for the head of the SSU, EMD-13477 for the body of the SSU and EMD- EMD-13578 for the subtomogram averaging of the whole ribosome. The atomic models have been deposited under the accession 7PKT for the LSU and 7PKQ for the SSU.

- although the authors thoroughly present the overall architectural features of this structurally divergent ribosome, it is unclear how the fragmented ribosomal RNA and perhaps newly recruited ribosomal proteins affect the functional sites, such as the decoding center and the peptidyl transferase center. This should be explicitly stated either in the results' Overall structure section or in the Discussion.

Although hinted in the text, the functional sites PTC (located on the L8 fragment) and the decoding center are almost unchanged compared to the "ancestral" bacterial ribosome. Especially, the PTC from H89 to H93 is particularly conserved, and located on one of the

largest fragments. This is a trend that is observed in every mitoribosomes described to date, as divergent as they can be, the functional centers are conserved. This observation is now reinforced in the text (lines 183-184 and 190-191).

- the authors should double check if the correct SSU maps have been used for calculating the map to model FSC in figure S2. The sudden drop in the plots may indicate the use of filtered maps.

The map to model FSC curves were calculated over the postprocessed and filtered maps. The drop is subsequent to the filtering up to the estimated resolution of each map. Using unfiltered maps may generate overfitting artifacts, as parts of the model may correlate with noise.

- the last two sentences of the second paragraph in the knock-down results chapter with references to figures 5A and 5B are duplicated.

Has been corrected.

- the authors established that the L2b RNA fragment is not a part of ribosomal RNA and is linked to mitochondrial genome maintenance, however, they should then elaborate more why the mS105 knockdown significantly lowers the relative L2b amounts.

This result is indeed intriguing. In this work, we were able to show that the L2b fragment is not part of the mitoribosome. We know that this fragment is transcribed and stably accumulates in *Chlamydomonas* mitochondria, however, its function remains entirely speculative. Vahrenholz et al. proposed that this fragment could be involved in the replication of the mitochondrial genome. Indeed, the end of the L2b rRNA sequence matches the outermost 86 nt of the two telomeres of the mitochondrial genome. In this hypothesis, the L2b fragment would prime the DNA synthesis at the genome's ends. However, for the moment, this remains to be demonstrated.

On the other hand, the presence of the mS105 protein, also called P32, is intriguing. The P32 protein is present in all eukaryotes and has been described as having several mitochondrial functions related to mitoribosome assembly. However, this is the first time that mS105/p32 has been identified as a core component of a ribosome. More in-depth analyses are needed to determine the role of this protein in mitoribosome assembly, translation, and/or replication. One way to determine if this effect is specific to the mS105 protein would be to get a mutant for another small subunit protein and see if we have the same effect. Still, the decrease of L2b rRNA level in the mS105 mutant strain might point out that this RNA function might be related to the function of mS105, and thus possibly to the mitoribosome or the SSU assembly/biogenesis or alternatively that mS105 is implicated in DNA maintenance by interacting with L2b. Additional discussion on the possible function of the L2b RNA is now included in the discussion section on lines 384-388.

- figures 6E and 6F should be larger.

Panels E, F and G of figure 6 are now larger.

Altogether, we wish to thank the reviewers for their constructive comments that have enabled us to improve our manuscript presentation and content.

We hope that our revised version of the manuscript will be acceptable for publication by Nature communications.

With best regards,

Philippe Giegé, Laurence Drouard and Yaser Hashem

Reviewers' Comments:

Reviewer #2:

Remarks to the Author:

Overall, I am satisfied with the changes to the text, methods, and figures presented in this revised manuscript. At this point, I have no further suggestions for improvement.

Reviewer #3:

Remarks to the Author:

The authors have adequately revised the manuscript in light of reviewers comments. The manuscript is therefore suitable for publication.